# BIRD: A Trustworthy Bayesian Inference Framework for Large Language Models

**Yu Feng**[1]* **Ben Zhou**[2] **Weidong Lin**[1] **Dan Roth**[1]
[1]University of Pennsylvania [2]Arizona State University

## Abstract

Predictive models often need to work with incomplete information in real-world tasks. Consequently, they must provide reliable probability or confidence estimation, especially in large-scale decision-making and planning tasks. Current large language models (LLMs) are insufficient for accurate estimations, but they can generate relevant factors that may affect the probabilities, produce coarse-grained probabilities when the information is more complete, and help determine which factors are relevant to specific downstream contexts. In this paper, we make use of these capabilities of LLMs to provide a significantly more accurate probabilistic estimation. We propose BIRD, a novel probabilistic inference framework that aligns a Bayesian network with LLM abductions and then estimates more accurate probabilities in a deduction step. We show BIRD provides reliable probability estimations that are 30% better than those provided directly by LLM baselines. These estimates further contribute to better and more trustworthy decision making.

## 1 Introduction

Accurate, interpretable, and unbiased probabilistic estimation is crucial to artificial intelligence (AI) systems in mission-critical tasks that require trustworthy decision making. This is because, in many real-world applications, AI systems have to work with partial and incomplete observations that are insufficient for deductive and deterministic decisions (McCarthy & Hayes, 1981). Consider the charging station planning scenario in Fig. 1, where we want to use large language models to decide where to build a certain number of charging stations over a region, subject to some constraints. Since we cannot always observe all the necessary information regarding a potential location, we can only provide probability or confidence estimates on whether we should use the candidate location. If the number of planned stations is significant, we need to perform an optimization process such as ILP (Roth & Yih, 2005) that maximizes the estimated probabilities of each individual station. A simple ranking of the candidates would not be sufficient in this scenario, as it may yield a suboptimal solution illustrated in Fig. 1. However, current LLMs are not ideal for such probability estimation tasks since 1) LLMs tend to be inaccurate and overconfident when estimating numerical confidences and probabilities (Xiong et al., 2024) and 2) they cannot provide an out-of-the-box interpretable and controllable process on how their estimates are derived (Li et al., 2024). This prevents us from using LLMs' decision-making capabilities in large-scale automatic decision-making tasks. Consequently, there is a need to find ways to infer better probabilities from these models.

In this work, we propose BIRD (**B**ayesian **I**nference f**r**om Abduction and **D**eduction). BIRD builds on three observations. The first observation is that although LLMs cannot produce convincing numerical probabilities directly, they can generate neutral and comprehensive real-world factors relevant to these probabilistic estimates. Moreover, we observe that they can make relatively correct coarse probability estimations when using these factors as contexts that provide more complete information. The last observation is that LLMs can decide how specific contexts and conditions are related to these factors via entailment classifications. Together, these observations motivate BIRD, where we approximate a Bayesian network (Friedman et al., 1997) using LLM-generated factors, optimize its internal conditional probabilities by sampling LLM coarse predictions on combinations of values of all the factors, and infer more accurate probabilities for specific downstream conditions by leveraging LLMs to map conditions to factors.

---

*fengyu1@seas.upenn.edu. Code available at https://github.com/CogComp/BIRD.

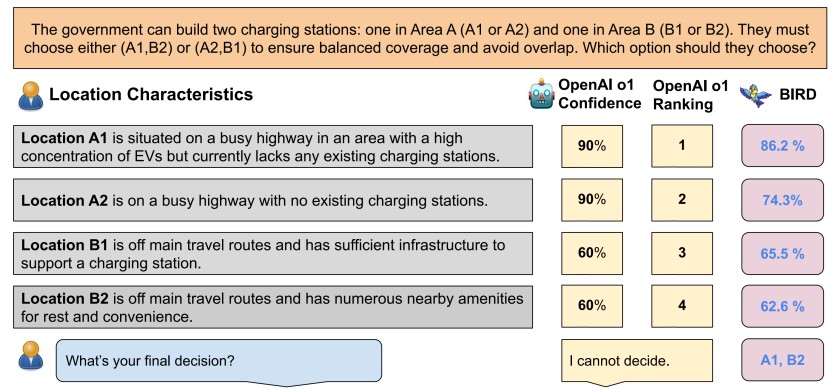

Figure 1: An example of mission-critical tasks. We first ask OpenAI o1 to separately predict the probability of building a charging station at each specific location. It estimates the same probabilities for two different conditions twice, while BIRD (ours) successfully distinguishes the difference and can thus help the user make a more informed decision. This further demonstrates that while LLMs are capable of coarse estimations, they struggle to generate accurate probabilities. We further show through OpenAI o1 ranking that although it can successfully rank all four locations based on how likely it should be used to build a new charging station, it is still insufficient to complete the task because of ties, i.e., (1,4) and (2,3) are the same from ranking perspectives.

As illustrated in Fig.2, we formulate the probability estimation problem as deciding $\mathbb{P}(O_i|C)$, where $O_i$ is a potential outcome (e.g., the government should/should not build a charging station here) and $C$ is the downstream condition or context (e.g., the location is on a busy highway) under a decision-making scenario $S$ (e.g., deciding whether to build a charging station at the location). Given $S$, BIRD first generates a comprehensive set of real-world factors and values $f_1, f_2, ..., f_n$ that may contribute to the probabilistic estimations. Then, it approximates a Bayesian network structure using the generated factors and the outcomes $\{O\}$. We propose a novel method for estimating the conditional probability table (CPT) for the outcome, i.e., $\mathbb{P}(O_i|f), f \in \mathcal{F}$ where $\mathcal{F}$ is the product space of value sets for all factors.[1] Specifically, we assume a set of learnable parameters $\mathbb{P}(O_i|f_j)$ representing the conditional probability of each possible outcome given a single variable $f_j$. We then sample $f \in \mathcal{F}$ and acquire LLM's coarse probability estimations $\mathbb{P}_{LLM}(O_i|f)$. Next, we approximate the conditional probability $\mathbb{P}_{estimated}(O_i|f)$ in the CPT using the learnable parameters $\mathbb{P}(O_i|f_j)$ based on a derived approximation formula and optimize these parameters by minimizing the distributional distance between $\mathbb{P}_{estimated}(O_i|f)$ and the LLM's coarse predicted probability $\mathbb{P}_{LLM}(O_i|f)$. From a high level, this abduction step aligns language models' internal probability estimation with Bayesian constraints and has better error tolerance. The resulting Bayesian network is generic to all possible conditions in the scenario $S$. In the following deduction step, we use the trained Bayesian parameters to estimate the probability of any possible conditions and observations $\mathbb{P}(O_i|C)$. To do this, we employ an entailment process to determine $\mathbb{P}(f_i|C)$, as shown by "LLM Entailment" in Fig. 2, and the outcome we are interested in, $\mathbb{P}(O_i|C) = \sum_{f \in \mathcal{F}} \mathbb{P}(O_i|f)\mathbb{P}(f|C)$.

As we show in §4, the inferred probabilities from BIRD are more reliable, consistent with human judgments on 30% more instances than those directly predicted by LLMs. We further show that BIRD probability predictions are good enough to be directly used for decision making (i.e., by computing an argmax on the predicted probabilities of each label for inference), outperforming standard chain-of-thought (CoT) (Wei et al., 2022) on several reasoning benchmarks. More importantly, BIRD achieves better probability estimations by optimizing an interpretable Bayesian network with natural-language-based variables. There are two additional benefits of BIRD. First, we show that BIRD-aligned probabilities can serve as more accurate and fine-grained supervision signals for training smaller models. In addition, we demonstrate that BIRD's factor generation process is neutral and unbiased to the specific conditions, so it can be used to generate better follow-up questions to improve decision-making confidence. Combining these findings, we show that BIRD improves trustworthiness — a crucial ability for mission-critical applications such as medical chatbots, where we need reliable decisions supported with enough evidence for high-confidence predictions.

---

[1]$\mathcal{F}$ is the set of all possible combinations of different assignments of values to all factors (e.g., selecting a value from each of the factors a/b/c/d/e/f in Fig. 2). Refer to §3.1.

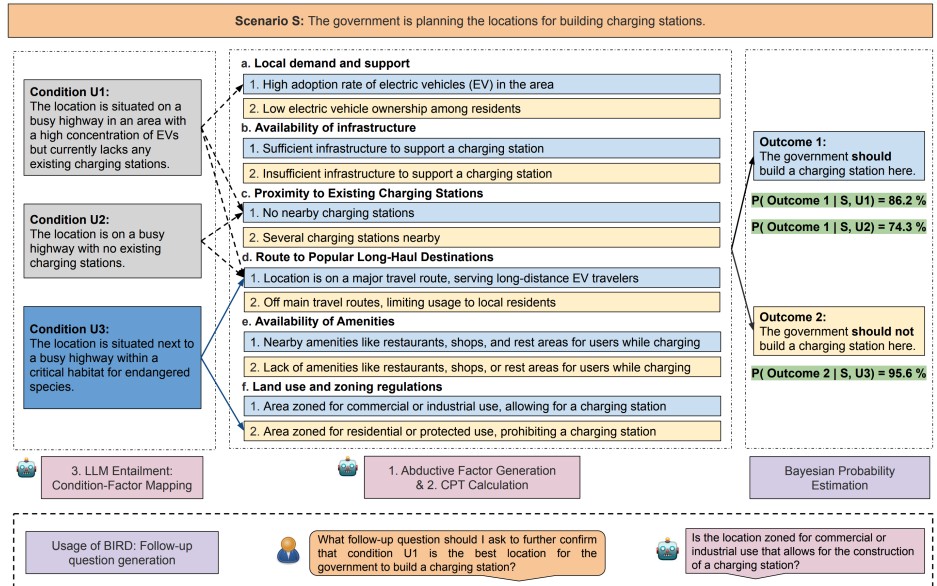

Figure 2: Overview of BIRD. Given a scenario $S$, LLMs generate the factors $F$ (a/b/...) with potential values ($f_1 \in \{\text{a.1,a.2}\}$). BIRD approximates a Bayesian network parameterized by $\mathbb{P}(O_i|f_j)$, and optimizes by sampling LLM coarse predictions on $\mathbb{P}_{\text{LLM}}(O_i|f), f \in \mathcal{F}$ ($\mathcal{F}$ is the set of all value combinations of $F$, e.g., $f = (\text{a.1}, \text{b.2}, \text{c.1}, \text{d.2}, \text{e.1}, \text{f.2})$), and minimizing the distributional distance between an approximated $\mathbb{P}_{\text{estimated}}(O_i|f)$ and $\mathbb{P}_{\text{LLM}}(O_i|f)$. At inference time, each context $C$ (S+U1/S+U2/...) is mapped to some $f_j$ via entailment, and a probability is derived using $\mathbb{P}_{\text{estimated}}$. BIRD can further generate follow-up questions.

## 2 RELATED WORK

**Direct Inference in LLMs.** Our paper is first related to LLM direct inference methods (Wei et al., 2022; Wang et al., 2023), and decomposition-based reasoning (Wolfson et al., 2020; Tafjord et al., 2022; Zhou et al., 2022; Kassner et al., 2023). Compared to these works, BIRD considers using abduction and deduction to facilitate probabilistic induction, which aligns with the slow-thinking systems (McGlynn, 2014; Lin et al., 2023). Our work is also related to chain-of-thought via latent-variable inference. Given a question-answer pair $(X, Y)$, existing methods (Hoffman et al., 2023; Hu et al., 2024) aim to find latent chains of thought – token sequences $Z$ that contribute the most to the conditional likelihood. While BIRD has a more structured $Z$ and focuses more on the proper probability estimation instead of merely finding the optimal $Z$.

**Uncertainty Estimations and Decision making under Uncertainty in LLMs.** Most current works focus more on sampling to access the model's uncertainty of factual knowledge (Kuhn et al., 2023; Yadkori et al., 2024; Lin et al., 2024). Some works verbalize the uncertainty of LLMs (Tian et al., 2023; Xiong et al., 2024). However, Xiong et al. (2024) shows that LLMs are often overconfident when directly verbalizing their confidence. Hou et al. (2024) is most related to us, as it also incorporates the intuition of decomposition, but it focuses more on addressing aleatoric uncertainty due to input ambiguity without rigorous probability calculation. This work is also related to decision making with uncertainty (Ren et al., 2023; Han et al., 2024; Liu et al., 2025) while all methods use LLM's direct uncertainty estimation such as token logits or verbalization. BIRD introduces external reliable Bayesian modeling to more accurately calibrate uncertainty in LLMs for decision making.

**Probabilistic Inference with LMs.** Chen et al. (2020) is one of the first papers to estimate a probability out of an NLI problem while with no explanation. Ozturkler et al. (2023) follows a similar intuition to ours, where the model first thinks by retrieving relevant associations, and then conducts probabilistic reasoning. Lew et al. (2020); Wong et al. (2023) discuss translating from natural language to the probabilistic language of thought. However, none of the papers tackle the real-world complex settings under incomplete information as comprehensively as we do with the level of interpretability to users of any level.

## 3 MODELING

### 3.1 BAYESIAN INFERENCE FROM ABDUCTION AND DEDUCTION

On a high level, given a decision $Y$ and partial information $X$, instead of direct inference using LLMs (induction : $X \rightarrow Y$), BIRD conceptualizes the general scenario behind $X$ into intermediate factors (abduction : $X \rightarrow Z$), and then fits a Bayesian model with $Z$. The model then estimates probabilities for $Y$ (deduction : $X, Z \rightarrow Y$). Our problem setting is a generic contextual inference problem of the type inspired by Feng et al. (2023). Given the context $C$, which includes a general scenario $S$ and a corresponding downstream condition $U$, our task is to estimate $\mathbb{P}(O_i|C), i = 1, 2, C = (S, U)$, where $\{O\}$ is denoted as $O_1$ and $O_2$. We decompose the overall context $C$ into the general scenario $S$ and downstream conditions $U$ because we want to build a neutral intermediate space $Z$ based on only $S$, so that the probability estimation is unbiased towards any specific conditions. This would mitigate models relying on incorrect signals, such as *charging stations are normally built in higher-income neighborhoods*, but more generic factors like *accessibility* for trustworthy decision making.

Instead of directly making a prediction, we approximate a controllable and interpretable inference process based on explicit Bayesian modeling (Gelman et al., 1995) containing natural-language-based variables. For a scenario $S$, we conceptualize $N$ factors $\{F_j\}_{j=1}^{N}$ based on $S$ that are relevant to the outcome $O$, which can take on two possible values, $O_i, i = 1, 2$, similar to the structure found in Bayesian networks (Koller & Friedman, 2009). Here the real-world factors and the outcome are equivalent to the variables in the network. Therefore, the choice between outcomes $O_1$ and $O_2$ depends on the factors, with each discrete factor $F_j$ having a set of possible values as $\mathcal{F}_j$, containing all possible details of the factor required for deciding between $O_1$ and $O_2$. We use $\mathcal{F} = \prod_{j=1}^{N} \mathcal{F}_j$ to denote the product space of value sets for all factors. For an instance in the space, each factor $F_j$ should be attributed a specific value $f_j$. We assume such derived factors are complete at this stage. That is, they include all the possible individual factors/angles that may affect the distribution of the outcomes.[2] As a result, $\mathcal{F}$ is a complete information space for decisions. As in Fig. 2, (factor a value 1 $a_1$, factor b value 1 $b_1$, factor c value 1 $c_1$, factor d value 1 $d_1$, factor e value 1 $e_1$, factor f value 1 $f_1$) is an element $f$ in the product space $\mathcal{F}$ where there are $2^6$ elements in total. We then calculate the conditional probability table (CPT) as $\mathbb{P}(O_i|f), \forall f \in \mathcal{F}, f = (f_1, f_2, ..., f_N)$.

We assume 1) $\{F_j\}_{j=1}^{N}$ is complete, so $O$ does not depend on any other intermediate factors, and 2) only one value can be assigned for each factor for a given $C$. The predictive probability for the outcome is thus obtained by marginalizing over the product space $\mathcal{F}$ for a context $C$ (details in §A.1):

$$\mathbb{P}(O_i|C) = \sum_{f \in \mathcal{F}} \mathbb{P}(O_i|f)\mathbb{P}(f|C) \tag{1}$$

where $i = 1, 2, C = (S, U), \forall f \in \mathcal{F}, f = (f_1, f_2, ..., f_N), f_j \in \mathcal{F}_j, j = 1, ..., N$. $\mathcal{F}$ is the product space of value sets for all factors. $f$ is a possible instance of complete information in the space.

**Assumption on Conditional Independence of Variables.** To provide relaxation for real-world applications, and since the factors should cover as many different aspects as possible, we assume factors are conditionally independent given the context. Therefore,

$$\mathbb{P}(O_i|C) = \sum_{f \in \mathcal{F}} \mathbb{P}(O_i|f) \prod_{j=1}^{N} \mathbb{P}(f_j|C) \tag{2}$$

In the following sections, we describe the missing implementation details of the framework: (1) the generation of the factors $\{F_j\}_{j=1}^{N}$ (§3.2); (2) the assignment of the conditional probability table $\mathbb{P}(O_i|f)$ (§3.3), and (3) the computation of observations $\mathbb{P}(f_j|C), j = 1, ..., N$ (§3.4).

### 3.2 ABDUCTIVE FACTOR GENERATION

Large language models, although they cannot solve some complicated cases, are shown to possess parametric knowledge that enables them to solve more common cases that can be directly found

---

[2]Note that our framework can later accommodate any additional factors by selectively recalculating specific parts as needed.

in their memory (Kambhampati et al., 2024; Li et al., 2024; Zhou et al., 2024). Our method for generating the factors follows this similar intuition. For a scenario $S$, we first do abstractions with LLMs to derive $N$ factors $\{F_j\}_{j=1}^N$. The most straightforward approach involves utilizing LLMs to directly obtain a list of factors and their potential values. However, direct generation is effective but not sufficiently comprehensive, and we want to fully utilize past similar occurrences that the model memorizes in their parametric knowledge.

Therefore, to encourage models to fully explore the range of alternatives, we divide the task into two stages: 1) generating sentences describing familiar situations that would increase the likelihood of an outcome, and 2) summarizing these sentences into factors with corresponding values. We show the prompts in Appendix Fig. 5 and Fig. 6. We further utilize LLMs to perform classification to assess how the value of each generated factor affect the decision, i.e., deciding the value supports which outcome, so that we can filter out factors that are unrelated to the outcomes. The prompt is in Appendix Fig. 7. In the example depicted in Fig. 2, a factor like "The color of the vehicle" will be eliminated since all potential values of this factor consistently support the same outcome, i.e., neutral.

### 3.3 ASSIGNMENT OF $\mathbb{P}(O_i|f)$ AS A CONSTRAINED OPTIMIZATION PROBLEM

We now calculate each $\mathbb{P}(O_i|f), i = 1, 2, \forall f \in \mathcal{F}$ in the conditional probability table of the Bayesian network. Since we do not have any observed data for approximation and LLMs alone cannot directly give an accurate assignment on $\mathbb{P}(O_i|f)$, we adopt the formula proposed in Bordley (1982), which derived a version of the logarithmic opinion pool using axioms from the theory of additive conjoint measurement. This formula is proved to be the only continuous (over the open interval (0, 1)) group probability formula under the assumption detailed in §A.2 that the decision maker has an intuitive weak ordering on the set of factor odds ratios and a "noninteraction" property of factors. Our setting fits the assumption as the weak ordering of decisions is the most fundamental condition of consistency in decision making and we assume the conditional independence of factors. Therefore,

$$\mathbb{P}(O_i|f) = \frac{\prod_{j=1}^N (\frac{\mathbb{P}(O_i|f_j)}{\mathbb{P}(O_i)})^{w_j} \mathbb{P}(O_i)}{\prod_{j=1}^N (\frac{\mathbb{P}(O_i|f_j)}{\mathbb{P}(O_i)})^{w_j} \mathbb{P}(O_i) + \prod_{j=1}^N (\frac{1-\mathbb{P}(O_i|f_j)}{1-\mathbb{P}(O_i)})^{w_j} (1 - \mathbb{P}(O_i))} \quad (3)$$

To determine the weights $w_j$,[3] , we adopt the second case scenario mentioned in Bordley (1982), i.e., $\sum_{j=1}^N w_j > 1$. In this case, the decision maker with complete information is more certain than the decision maker with only one piece of information through acquiring all different pieces of information from different factors. Since there isn't a priori information that suggests another choice exists, we use equal weight $w_j = 1, j = 1, ..., N$ and assume $\mathbb{P}(O_i) = 50\%, i = 1, 2$. The final estimation formula can be rewritten as:

$$\mathbb{P}_{\text{estimated}}(O_i|f) = \mathbb{P}_{\text{estimated}}(O_i|f_1, f_2, ...f_N) \approx \frac{\prod_{j=1}^N \mathbb{P}(O_i|f_j)}{\prod_{j=1}^N \mathbb{P}(O_i|f_j) + \prod_{j=1}^N (1 - \mathbb{P}(O_i|f_j))} \quad (4)$$

**Constrained optimization for estimating $\mathbb{P}(O_i|f)$.** From Eq. 4, it is not hard to see that each $\mathbb{P}(O_i|f), \forall f \in \mathcal{F}$ can now be approximated using only $\mathbb{P}(O_i|f_j), \forall f_j \in \mathcal{F}_j, j = 1, ..., N$, which we view as learnable parameters. As discussed in §1, LLMs can perform relatively correct coarse estimation under complete information, which refers to contexts with all factors' values present (i.e. $f \in \mathcal{F}$). Consequently, we prompt the LLMs to output verbalized probability estimations for outcome $O_i$ given $f$. This approximates LLM's predictions of $\mathbb{P}_{LLM}(O_i|f)$. With $\mathbb{P}_{\text{estimated}}$ and $\mathbb{P}_{LLM}$, we can now form a constraint optimization problem that finds the values of parameters $\mathbb{P}(O_i|f_j)$ that minimize the distributional distance between the Bayesian estimated probabilities $\mathbb{P}_{\text{estimated}}(O_i|f)$ and LLM estimated probabilities $\mathbb{P}_{LLM}(O_i|f), \forall f \in \mathcal{F}$, while conforming to the structure in Eq. 4 and preliminary assessments about each $\mathbb{P}(O_i|f_j), \forall f_j \in \mathcal{F}_j, j = 1, ..., N$ provided by the LLM classification as mentioned in §3.2.

**Learning algorithm for constrained optimization to estimate $\mathbb{P}(O_i|f)$.** We propose a gradient-descent algorithm for this optimization problem. The detailed algorithm can be found in §A.3, and

---

[3]Weights here refer to a set of "hyperparameters" that decides how highly the final decision regards each factor and how correlated the factors are considered, not the learnable parameters in our Bayesian network.

we give a general description here. As the parameter initialization, we utilize the LLM classification as mentioned in §3.2, where we direct the LLMs to ascertain which outcome is more likely supported by the value of each factor:

$$\mathbb{P}_{\text{init}}(O_i|f_j) = \begin{cases} 75\% & f_j \text{ supports outcome } i \\ 50\% & f_j \text{ is neutral} \\ 25\% & f_j \text{ supports opposite outcome } \neg i \end{cases} \tag{5}$$

For the training data, given an instance of complete information $f' = (f'_1, f'_2, ..., f'_N)$, we prompt LLM to output a verbalized probability score for each outcome $\in$ {very unlikely, unlikely, somewhat unlikely, neutral, somewhat likely, likely, very likely} which we use as $\mathbb{P}_{LLM}(O_i|f')$. The detailed prompt can be found in Appendix Fig. 10. We then assign a specific probability value for each coarse estimation, i.e., very unlikely: 0%, unlikely: 20%, somewhat unlikely: 40%, neutral: 50%, somewhat likely: 60%, likely: 80%, very likely: 100%, as supervision signals to learn the correct value of $\mathbb{P}(O_i|f'_j), j = 1, ..., N$ for each $f'_j$ in $f'$ using Eq.4. We randomly sample 128 instances in the space of complete information as the training data for each scenario. Note that we manually assign probability values for LLM coarse decisions, but they can be changed as hyperparameters.

We adopt two loss functions: 1) we use the regular mean squared error loss for optimization and denote the loss as $\ell_{\text{MSE}}(\mathbb{P}_{\text{estimated}}(O_i|f'), \mathbb{P}_{\text{LLM}}(O_i|f'))$; 2) we use the margin ranking loss to ensure the final trained individual probability preserve the original direction of its supported outcome and denote the loss as $\ell_{MR}$:

$$\ell_{\text{MR}} = \max(0, -y_{\text{target}}(O_i|f_j)(\mathbb{P}_{\text{trained}}(O_i|f_j) - 0.5) + \epsilon)$$
$$y_{\text{target}}(O_i|f_j) = \text{sgn}(\mathbb{P}_{\text{init}}(O_i|f_j) - 50\%) \tag{6}$$
$$\mathbb{P}_{\text{trained}}(O_i|f_j) = \sum_{f^* \in \mathcal{F}^*} \mathbb{P}(O_i|f^* \cup f_j)\mathbb{P}(f^*|f_j) = \frac{1}{\text{card}(\mathcal{F}^*)} \sum_{f^* \in \mathcal{F}^*} \mathbb{P}(O_i|f^* \cup f_j)$$

$\forall f_j \in \mathcal{F}_j, j = 1, ..., N$. We apply probability marginalization to get the trained individual probability and approximate it with the mean values since there are no explicit priors. We use $\mathcal{F}^*$ to denote the product space of the remaining factors except $\mathcal{F}_j$, which is a subspace of $\mathcal{F}$ and $f^* = (f_1, f_2, ..., f_{j-1}, f_{j+1}, ..., f_N), f^* \in \mathcal{F}^*$. Therefore, the final loss function we use for training is defined as follows:

$$\mathcal{L} = \ell_{\text{MSE}} + \alpha\ell_{\text{MR}} \tag{7}$$

where $\alpha$ balances the two losses. We use SGD as the optimization function. As a result, we can estimate any $\mathbb{P}(O_i|f), \forall f \in \mathcal{F}$ with the learned $\mathbb{P}(O_i|f_j), \forall f_j \in \mathcal{F}_j$ with Eq.4.

**Customization to human preference.** A benefit of our design is that we can explicitly ask for human preferences on how a particular factor value will affect the outcome, i.e., manually setting $\mathbb{P}(O_i|f_j)$. Such preferences are desirable in certain applications with known constraints, and they can be easily integrated in Eq.4 by replacing $\mathbb{P}(O_i|f_j)$ with the human preferred value.

### 3.4 COMPUTATION OF $\mathbb{P}(f_j|C)$ WITH LLM ENTAILMENT

This section discusses how we compute $\mathbb{P}(f_j|C)$. As noted in §3.1, $C$ is the context containing the general scenario $S$, and a specific additional condition $U$. Given $C$, we employ an entailment task formulation to find which factors and their corresponding values are implied or mapped by $C$. Specifically, we adopt the prompt in Appendix Fig. 8 where we directly ask if the context entails a value from a factor. We conduct another round of self-correction using the prompt in Appendix Fig. 9 to check if the model believes all the previously identified values are actually entailed.

We define the notion of observed factors, which refers to a subset of factors among all factors that are being implied by the context. For example, as shown in Fig. 2, context $S + U_1$ implies $a_1, c_1$, and $d_1$, $S + U_2$ implies $c_1$ and $d_1$, while the additional condition $S + U_3$ implies $d_1$ and $f_2$. We assume only one value can be chosen per factor. We write the product value space of the implied factors as $\mathcal{F}' = \prod_{m=1}^{M} \mathcal{F}_{j_m}$, which is a subspace of $\mathcal{F}$. We define the conditional probabilities of the values of these observed factors to be 1 if the value is implied by $C$, a 0 if not:

$$\mathbb{P}(f_{j_m}|C) = \begin{cases} 1 & f_{j_m} = f^*_{j_m}, f_{j_m} \in \mathcal{F}_{j_m} \\ 0 & f_{j_m} \neq f^*_{j_m}, f_{j_m} \in \mathcal{F}_{j_m} \end{cases} \tag{8}$$

Where the implied $f_{j_m}^* \in \mathcal{F}_{j_m}, \forall m = 1, ..., M$. For the unobserved factors, we have no information regarding which value to be selected. In order to be neutral and unbiased, we assume that, in an unobserved factor, each value has an equal probability of being selected. We use $F'^{\perp} = \prod_{m=M+1}^{N} \mathcal{F}_{j_m}$ to denote the space of unobserved factors, which is the orthogonal complement space of $\mathcal{F}'$ in $\mathcal{F}$,

$$\mathbb{P}(f_{j_m}|C) = \frac{1}{\text{card}(\mathcal{F}_{j_m})}, f_{j_m} \in \mathcal{F}_{j_m} \tag{9}$$

Where $\forall m = M + 1, ..., N$. We can therefore calculate the final probability as presented in Eq.1, utilizing the formulations provided in Eqs.2 through 9.

## 4 EXPERIMENTS

### 4.1 DATASETS AND SETTINGS

We convert three datasets focusing on reasoning and planning to our problem setting including $O$, $S$, and $U$, as described in §3.1. We detail this process in §A.4. COMMON2SENSE (Singh et al., 2021) focuses on commonsense reasoning, and we gather 216 scenarios and 3822 instances, where 9 additional conditions on average support each outcome. PLASMA (Brahman et al., 2023) focuses on planning, and we gather 279 scenarios and 1395 instances, where 5 additional conditions support the less common outcome. TODAY (Feng et al., 2023) focuses on temporal reasoning, where there are 1000 instances. Among the three datasets, the most challenging task is TODAY and the easiest is COMMON2SENSE, judged by baseline model performances. The complexity of the scenario and conditions is highest in TODAY. The conditions support the less common outcome in PLASMA and COMMON2SENSE presents general decision making with common sense.

We conduct our experiments with Llama-2-70b-instruct and Llama-3.1-70b-Instruct.[4] For the factor generation task in §3.2, we generate 10 sentences for each outcome before summarization. For the LLM classification task for factor-outcome mapping, we generate 3 times and conduct a majority vote. For the LLM entailment task in §3.4, we sample 3 times and conduct a majority vote with 1 round of self-reflection. We set learning rate = 1e-02, epoch = 20, batch size = 4, $\epsilon = 0$, $\alpha = 10$ for the learning algorithm mentioned in §3.3.

### 4.2 INTRINSIC EVALUATIONS OF THE BIRD FRAMEWORK

**Reliability of BIRD's Estimated Probabilities.** We first propose an evaluation method that directly benchmarks the quality of probability estimations. We adopt a similar setting as mentioned in §3.1. Given a scenario $S$ and two potential outcomes $O_1$ and $O_2$, we find two specific additional conditions $U_1$ and $U_2$, both supporting $O_1$ over $O_2$.[5] $S + U_1$ and $S + U2$ will form two contexts $C_1$ and $C_2$, respectively. The goal of this task is to evaluate if models can assign fine-grained probability estimations, that is, sufficient to distinguish which context between $C_1$ and $C_2$ *better* supports $O_1$ over $O_2$ or if both $C_1$ and $C_2$ *equally* support $O_1$ over $O_2$, by predicting probability values of $P(O_i|C1)$ for $C_1$ and $P(O_i|C2)$ for $C_2$ *individually*.[6] That is, assuming $C_1$ better support $O_1$ over $O_2$, we expect $P(O_1|C_1) > P(O_1|C_2) > P(O_2|C_2) > P(O_2|C_1)$, and vice versa. If $C_1$ and $C_2$ equally support $O_1$ over $O_2$, we should see $P(O_1|C_1) = P(O_1|C_2) > P(O_2|C_1) = P(O_2|C_2)$. This task is significantly more difficult than providing two contexts that support different outcomes, as systems will have to understand the nuances in probability estimation.

We collect human gold labels for which context better supports the gold outcome with rigorous quality control using MTurk. We explain our choice of this relative evaluation over asking human annotators to provide absolute probabilities from a psychological perspective, along with the details of human annotation in §A.5. We evaluate on 350 randomly sampled instances from COMMON2SENSE when the conditions can be mapped to at least one factor and therefore BIRD does not produce "unknown"[7].

---

[4]https://huggingface.co/upstage/Llama-2-70b-instruct and https://huggingface.co/meta-llama/Llama-3.1-70B-Instruct

[5]Here $O_1$ will be the gold outcome. During evaluation, the two conditions can also support $O_2$ over $O_1$.

[6]For proper evaluation, systems predict probabilities one at a time, without seeing other contexts.

[7]Although the setting may seem to favor our proposed method, it is a fair comparison, because "unknown" predictions are label-agnostic, and we can use the same direct inference approach, such as CoT, for such cases.

| Model | Different$_1$ | Different$_2$ | Same | Average |
|---|---|---|---|---|
| Random Guessing | 0.333 | 0.333 | 0.333 | 0.333 |
| GPT3.5 CoT | 0.306 | 0.306 | 0.242 | 0.283 |
| GPT4 CoT | 0.312 | 0.357 | 0.216 | 0.289 |
| Llama2-70b Instruct Logit | 0.263 | 0.228 | 0.205 | 0.228 |
| Llama2-70b Instruct Vanilla | 0.375 | 0.333 | 0.243 | 0.311 |
| Llama2-70b Instruct CoT | 0.315 | 0.323 | 0.254 | 0.294 |
| Llama3.1-70b Instruct Logit | 0.300 | 0.282 | 0.242 | 0.269 |
| Llama3.1-70b Instruct Vanilla | 0.365 | 0.301 | 0.251 | 0.303 |
| Llama3.1-70b Instruct CoT | 0.373 | 0.351 | 0.187 | 0.309 |
| Llama2-70b Instruct EC* | 0.530 | 0.529 | 0.207 | 0.503 |
| Llama3.1-70b Instruct EC* | 0.535 | 0.538 | 0.286 | 0.511 |
| GPT4 EC* | 0.588 | 0.533 | 0.300 | 0.540 |
| Llama3.1 BIRD (*ablation w 1/2 assumption*) | 0.527 | 0.532 | 0.196 | 0.480 |
| Llama3.1 BIRD (*ablation w 1/n assumption*) | 0.572 | 0.584 | 0.272 | 0.532 |
| Llama3.1 BIRD (*ablation w fixed initial prob*) | 0.614 | 0.597 | 0.337 | 0.568 |
| Llama2 BIRD (*ours w optimized prob*) | **0.614** | 0.624 | **0.450** | **0.592** |
| Llama3.1 BIRD (*ours w optimized prob*) | 0.612 | **0.625** | 0.382 | 0.588 |

Table 1: Preference-based pairwise evaluation using F1 score for predicted probabilities across various model settings. Different$_1$ indicates the F1 score for a category where $C_1$ supports the gold outcome better than $C_2$, and vice versa for Different$_2$. Same refers to the F1 score of the category where $C_1$ and $C_2$ equally support the outcome. Average refers to calculating F1 metrics globally by counting the total true positives, false negatives, and false positives. EC* settings see both $C_1$ and $C_2$ at the same time instead of predicting individual probabilities, hence not directly comparable.

We report the F1 score as the evaluation metric. We further explain how the F1 score is computed in §A.5.

We compare the following methods that directly acquire probability estimations from LLMs as baselines. 1) **Vanilla** (Wang et al., 2023): models directly verbalize the estimated probabilities. 2) **Logit**: we select the normalized token probability of the decision. 3) **CoT** (Wei et al., 2022): models output a reasoning process followed by verbalized estimated probabilities. 4) **EC**: models select which additional condition better supports the outcome without outputting any probabilities. Note that this formulation is unfair to BIRD and other baselines that individually estimate the probabilities. To demonstrate the effectiveness of our learning algorithm in §3.3, we propose two ablation baselines that use other methods to find the conditional probabilities in the Bayesian network: 1) **1/2 and 1/N Assumptions**: We adopt two neutral assumptions without LLM classifications for CPT calculation. 1/2: $\mathbb{P}(O_i|f)$ is 1 only if all factors' values, except for any neutral ones, support $O_i$; if not, 0. 1/N: $\mathbb{P}(O_i|f)$ is calculated as the percentage of factors whose values support outcome $O_i$. 2) **Fixed Initial Prob**: We utilize the initial probability of Eq.5 to directly estimate $\mathbb{P}(O_i|f)$ with Eq.4. The details of the baselines are presented also in §A.6.

Table 1 lists the results. Our framework exhibits superior alignment with human preference, with both models achieving an F1 of 59%, more than 30% higher than GPT-4. Such results demonstrate that BIRD produces reliable probability estimation without additional human supervision. Notably, such probability estimation is done by judging each condition separately (i.e., BIRD assigns one probability to condition 1, and later another probability to condition 2). We find that vanilla language models do much better when both conditions are provided at the same time (i.e., the EC setting) where the model can perform argmax operations. However, even under this setting that is unfair to our framework, BIRD still outperforms GPT-4 by 5%. All results suggest that vanilla LLMs cannot properly estimate probabilities, and BIRD effectively improves them. In addition, by outperforming all the ablation settings, we demonstrate that our constrained optimization algorithm proposed in §3.3 effectively approximates better Bayesian network's conditional probabilities.

**Applying BIRD's Probabilities in Decision Making.** In this section, we evaluate the performance of applying the predicted probabilities from BIRD in decision-making tasks. In the general decision-making setting, we similarly assume a context $C = S + U$, two possible outcomes $O_1$ and $O_2$,

| Dataset | Bird_1/2 | Bird_1/n | Bird_fixed_prob | Bird_optimized (ours) | CoT |
|---|---|---|---|---|---|
| *Llama-2-70b-instruct* | | | | | |
| Today | 73.7 | 72.8 | 73.4 | **73.9** | 71.5 |
| Plasma | 72.8 | 72.3 | 72.7 | 74.0 | **76.8** |
| Common2sense | 86.9 | 86.8 | 87.5 | 89.0 | **93.8** |
| *Llama-3.1-70b-Instruct* | | | | | |
| Today | 65.5 | 68.9 | 65.5 | **74.3** | 72.6 |
| Plasma | 71.3 | 66.5 | 65.7 | **73.0** | 71.5 |
| Common2sense | 78.1 | 85.4 | 86.7 | **92.3** | 90.8 |

Table 2: Performance comparisons of Bird and baselines on decision-making benchmarks. Bird_optimized is our final model; chain-of-thought with self-consistency (CoT) is our main baseline; others are ablation baselines.

and the task is to decide which outcome is more likely given $C$, similar to Fig. 2. The inference process we use for Bird is simply taking the argmax between $\mathbb{P}(O_1|C)$ and $\mathbb{P}(O_2|C)$. We remove all instances where they cannot be mapped to any factor values(i.e., Bird's decision is "unknown"). This does not undermine our experiment setting because such removal is label-agnostic, and we can always fall back to other methods such as CoT when Bird predicts "unknown". We compare our framework with chain-of-thought inference with self-consistency on the same instances where Bird predictions are not "unknown". We present the prompt for CoT in Appendix Fig.14.

The results are shown in Table 2. Our first observation is that Bird's probability estimations are accurate enough for direct decision making. Our proposed method outperforms the chain-of-thought baseline with Llama-3.1 on all benchmarks and performs comparably using Llama-2 as the base model. This is partially because Llama-3.1 generates more diverse and accurate factors, and we analyze this in §A.7. At the same time, we see that Bird performs better on the most challenging Today dataset on both models. This demonstrates that Bird is a desirable solution for more complex scenarios where CoT tends to fail more. We also observe the inferior performance of our method compared to CoT may be attributed to the fundamental differences between probability estimation and hard label classification. Table 6 in the appendix shows an example that CoT sometimes does correctly due to the spurious signals and biases in the conditions, which highlights the importance of our controllable inference approach. We can conclude that Bird outperforms direct inference methods like CoT in most cases and is at least comparable in decision making while offering significantly better interpretability and controllability throughout the process.

**Ablation Studies on Bird's Components.** We conduct an ablation study regarding factor generation. We compare the final decision-making accuracy based on our proposed abductive sampling method and the direct factor generation method mentioned in §3.2. Appendix Table 7 shows the results: we see that our proposed method, on average, surpasses the direct method by 4.4%/3.9% in accuracy for Llama-2/ Llama-3.1, indicating the effectiveness of our proposed factor generation that collects more comprehensive signals from models' parametric knowledge and memorization. We also conduct ablation studies on different ways of estimating the conditional probabilities of our Bayesian network (i.e., 1/2, 1/n, and fixed initial prob). Results in Tables 1 and 2 show that our proposed constrained optimization learning algorithm effectively finds more accurate estimations by outperforming all ablation baselines. We demonstrate that both losses in Eq.7 are crucial in §A.9.

### 4.3 Extrinsic Applications of the Bird Framework

**The Usage of the Reliably Estimated Probability.** As shown in §4.2, Bird can produce reliable probabilities that contribute to better decision making. Such findings motivate us to investigate whether these probabilities can be used as supervision signals, as models may benefit from seeing more than the discrete classification labels (Zhou et al., 2020). Specifically, we use Bird to estimate probabilities of some decision-making training instances and finetune a T5-large model using these generated probabilities. We then evaluate its performance on cross-domain datasets. We fully replicate the experimental settings described in Feng et al. (2023) and the details can be referred to in §A.10. We use instances from Common2sense, where the argmax of the estimated probabilities from

our framework using Llama-2 supports the correct outcome, as additional training data, besides the originally proposed training data in that paper. We test on two general temporal reasoning datasets, MATRES (Ning et al., 2018) and TRACIE (Zhou et al., 2021), as well as two commonsense reasoning and planning datasets, TODAY, and PLASMA. The results are shown in Table 3. We see that models achieve an average performance increase of 1.3% compared with BIRD-generated probabilities as additional signals. We use cross-entropy loss with class probabilities as the target for our soft probability setting and cross-entropy loss with class labels as the target for the hard-label setting. Such results suggest another effective application of the BIRD framework, where smaller models can benefit from soft probability alignment and make better overall predictions.

| Model (Train Data) | TODAY (exp) | TODAY | TRACIE | MATRES | PLASMA | Average |
|---|---|---|---|---|---|---|
| T5_large | 77.8 | 57.9 | **73.0** | **83.5** | 48.1 | 65.6 |
| + hard label | 83.6 | 62.9 | 68.2 | 75.2 | **65.2** | 67.9 |
| + BIRD prob (*ours*) | **84.3** | **63.4** | 71.4 | 77.6 | 63.3 | **68.9** |
| PatternTime | 82.2 | 60.9 | **79.8** | **85.8** | 50.5 | 69.3 |
| + hard label | 83.9 | 61.8 | 76.2 | 83.7 | 56.9 | 69.7 |
| + BIRD prob (*ours*) | **85.1** | **62.6** | 75.7 | 85.1 | **61.5** | **71.2** |

Table 3: System performances under different supervision data across three binary temporal benchmarks and one binary planning benchmark. For simplicity, we use "hard label" representing that we use COMMON2SENSE supervision data with explicit binary labels, and "BIRD prob" representing that we use COMMON2SENSE supervision data with estimated probabilities. TODAY (exp) uses gold explanations during evaluation.

**Using BIRD to Generate Better Follow-up Questions.** Generating accurate and informative follow-up questions is an important goal for LLMs to be applied in mission-critical tasks that require unambiguous decision making. A natural benefit of our BIRD framework is that it analyzes what needs to be known through the comprehensive and unbiased abduction step (i.e., factor generation) and interprets how new information will change the probability or confidence of a desired decision during the deduction step (i.e., probability estimation). We conduct a follow-up question generation experiment to demonstrate this benefit. For a given decision-making problem, we ask both BIRD and LLMs to generate binary (yes/no) follow-up questions. For BIRD, we identify implied factors, choose the most important unmentioned one, adopt the most important value aligning with the current outcome, and rephrase it as a question. We directly ask LLM to generate questions for better decision making. The details can be referred to in §A.11. We then recruit crowd-workers on MTurk to provide pairwise preference between the BIRD generated question and the question generated directly by the LLM. We do this on 250 examples from COMMON2SENSE. An example can be seen in Appendix Fig. 4 and the details can also be seen in §A.11. Appendix Table 9 shows that the follow-up questions generated by BIRD are more often preferred: questions generated by BIRD are preferred 52.8% of the time, while those directly generated by LLM are preferred 32.8% of the time; in the remaining 14.4%, both are considered equally useful. This result motivates future work on applying BIRD in an interactive agent system for more efficient and controlled decision making.

## 5 CONCLUSION

In this paper, we propose BIRD, a Bayesian inference framework for large language models. BIRD utilizes LLMs' existing reliable capabilities such as abductive thinking and coarse-grained classification to optimize an explicit Bayesian network for more reliable probability estimation. We show that BIRD can output accurate probabilities, outperforming LLM vanilla probabilties by 30% on direct estimation tasks, and outperform chain-of-thought on decision-making tasks. At the same time, BIRD's outputs are explainable, because of its natural-language-based variables in the fully-interpretable Bayesian network. We also show that BIRD can be used in several other settings, such as generating fine-grained supervision signals and asking informative follow-up questions, to further improve the trustworthiness and controllability of LLMs when applied to real-world applications.

## 6 ACKNOWLEDGEMENTS

This work was sponsored by the Army Research Office and was accomplished under Grant Number W911NF-20-1-0080. The views and conclusions contained in this document are those of the authors and should not be interpreted as representing the official policies, either expressed or implied, of the Army Research Office or the U.S. Government. The U.S. Government is authorized to reproduce and distribute reprints for Government purposes notwithstanding any copyright notation herein. This work was also supported by the Office of Naval Research [ONR N00014-23-1-2364]. We thank Kyurae Kim, Ollie Liu, Vivek Gupta and colleagues at CogComp UPenn for valuable discussions.

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

## A   APPENDIX

### A.1   DERIVATION OF THE EQUATION FOR UNCERTAINTY ESTIMATION

We derive the outcome probability for the given context $C$ of partial information, i.e., any additional condition $U$ under the scenario $S$, based on the complete information space $\mathcal{F}$ of scenario $S$:

$$
\begin{aligned}
\mathbb{P}(O_i|C) &= \sum_{f\in\mathcal{F}} \mathbb{P}(O_i|f,C)\mathbb{P}(f|C) \\
&= \sum_{f\in\mathcal{F}} \mathbb{P}(O_i|f,C)\mathbb{P}(f|C)(\mathbb{1}\{f\cap C = f\} + \mathbb{1}\{f\cap C \neq f\}) \\
&= \sum_{f\in\mathcal{F}\cap\{f\cap C=f\}} \mathbb{P}(O_i|f,C)\mathbb{P}(f|C) + \sum_{f\in\mathcal{F}\cap\{f\cap C\neq f\}} \mathbb{P}(O_i|f,C)\mathbb{P}(f|C) \\
&= \sum_{f\in\mathcal{F}\cap\{f\cap C=f\}} \mathbb{P}(O_i|f)\mathbb{P}(f|C) + \sum_{f\in\mathcal{F}\cap\{f\cap C\neq f\}} \mathbb{P}(O_i|f,C)\mathbb{P}(f|C) \\
&= \sum_{f\in\mathcal{F}\cap\{f\cap C=f\}} \mathbb{P}(O_i|f)\mathbb{P}(f|C) + \sum_{f\in\mathcal{F}\cap\{f\cap C\neq f\}} \mathbb{P}(O_i|f)\mathbb{P}(f|C) \\
&= \sum_{f\in\mathcal{F}} \mathbb{P}(O_i|f)\mathbb{P}(f|C)
\end{aligned}
\tag{10}
$$

Where the fourth line follows from that $\mathbb{P}(O_i|f,C) = \mathbb{P}(O_i|f)$ since $f\cap C = f$. And the second last line is derived from that $\mathbb{P}(f|C) = 0$ if $f\cap C \neq f$ as for a given $C$, only one value can be chosen for each factor as in Fig. 2.

### A.2   ASSUMPTIONS FOR ESTIMATING $\mathbb{P}(O_i|f)$

**Weak ordering of the set of factor odds ratios.**   Suppose given factors $f_1, f_2, ..., f_N$, we assess odds of $o_1, o_2, ..., o_N$, respectively in favor of the outcome $O_i, i = 1, 2$ occurring. $o_j = \frac{\mathbb{P}(O_i|f_j)}{1-\mathbb{P}(O_i|f_j)}, j = 1, ..., N$, the intuitive ordering $\geq_E$ is defined as:

$$
(o_1, o_2, ..., o_N) \geq_E (o_1^*, o_2^*, ..., o_N^*)
\tag{11}
$$

if and only if the decision maker feels at least as sure about the outcome $O_i$, occurring when given factors $f_1, f_2, ..., f_N$ we assess odds $o_1, o_2, ..., o_N$ -as when given factors $f_1, f_2, ..., f_N$ we assess odds $o_1^*, o_2^*, ..., o_N^*$. The ordering $\geq_E$ is a Weak Ordering if and only if for any $(o_1, o_2, ..., o_N), (o_1^*, o_2^*, ..., o_N^*), (o_1^{**}, o_2^{**}, ..., o_N^{**})$, we have:

- (a) either $(o_1, o_2, ..., o_N) \geq_E (o_1^*, o_2^*, ..., o_N^*)$ or $(o_1^*, o_2^*, ..., o_N^*) \geq_E (o_1, o_2, ..., o_N)$,
- (b) if $(o_1, o_2, ..., o_N) \geq_E (o_1^*, o_2^*, ..., o_N^*)$ and if $(o_1^*, o_2^*, ..., o_N^*) \geq_E (o_1^{**}, o_2^{**}, ..., o_N^{**})$ then $(o_1, o_2, ..., o_N) \geq_E (o_1^{**}, o_2^{**}, ..., o_N^{**})$.

**"Noninteraction" property of factors.**   The non-interaction condition says that the decision maker's interpretation of how one factor affects the outcome does not depend on how other factors affect the outcome.

### A.3   LEARNING ALGORITHM FOR ESTIMATING $\mathbb{P}(O_i|f)$

We present the complete algorithm in Appendix Algorithm 1. Since $\mathbb{P}(O_1)$ and $\mathbb{P}(O_2)$ are complementary, we only need to learn $\mathbb{P}(O_1|f_{ja}), f_{ja} \in \mathcal{F}_j, j = 1, ..., N, a = 1, ..., \mathrm{card}(\mathcal{F}_j)$, and $\mathbb{P}(O_2|f_{ja}) = 1 - \mathbb{P}(O_1|f_{ja})$. As a result, for a given scenario $S$, the number of learnable parameters is the total number of values for all the generated factors, i.e., $\sum_{j=1}^{N} \mathrm{card}(\mathcal{F}_j)$.

All training instances for a scenario are drawn from the complete information space for the same scenario $S$ and possible outcomes $\{O\}$ and will share the same factors $\{F_j\}_{j=1}^{N}$. We randomly sample 128 instances in the space of complete information as the training data for each scenario. This represents the minimum number of instances required for effective training. Ideally, increasing the number of sampled instances would improve the model's alignment with the underlying LLM. For a scenario with fewer than 128 instances in the space of complete information, we employ a resampling technique, selecting instances repeatedly until we reach a total of 128 samples.

The value of $\mathbb{P}_{\text{init}}(O_i|f_j)$ in Eq.5 is assigned as it represents random initialization. To determine the mappings of rankings to probabilities in the algorithm, we consulted two psychology experts and adopted the Likert scale theory. The finalized mapping between verbalized probabilities and numerical probabilities reflects typical human behavior and effectively distinguishes between verbalized probabilities in an unbiased manner. Additionally, slight adjustments to the mappings do not noticeably affect overall performance. All training hyperparameters for the algorithm are optimized using a grid search with a hold-out validation set from PLASMA.

### A.4 EXPERIMENT DATASET DESCRIPTION

We convert three datasets focusing on reasoning and planning to our problem setting.

- **COMMON2SENSE** (Singh et al., 2021): focuses on commonsense reasoning and planning. The original task constitutes a multi-domain complementary commonsense reasoning benchmark and comprises natural language true/false statements. We exclusively focus on instances of comparative reasoning where a pre-trained smaller model exhibits low confidence in its prediction logits. Using GPT-4[8], we rewrite each instance statement into its opposite, creating two distinct outcomes. We then outline a general scenario based on the two statements and generate 10 additional sentences for each statement, with detailed conditions that would support the statement. Additionally, we implement a reverse verification step to eliminate any incorrect additional sentences to ensure dataset quality. We gather 216 scenarios and 3822 instances. 9 additional conditions on average support each outcome.

- **TODAY** (Feng et al., 2023): focuses on temporal reasoning. The dataset is designed to study how introducing an extra sentence to a specific scenario affects the temporal relation, aligning well with the context of our task. There are 1000 instances.

- **PLASMA** (Brahman et al., 2023): focuses on planning. The original task is to generate or revise a plan based on a given goal while adhering to an additional condition. We've observed that the key variations between the initial and the revised plans given the additional condition typically focus on a specific step within the plan. We then utilize GPT-4 to identify the step and adapt the description of the step in the original and revised plans into two potential outcomes. We manually identify the less common outcome in the 279 scenarios and use GPT-4 to generate 5 additional sentences for the outcome, with detailed conditions that would support this outcome instead of the other. We gather 279 scenarios and 1395 instances with 5 additional conditions for each scenario.

For the two GPT-4 generated datasets, we recruited the authors and their colleagues to conduct a human quality check on 100 randomly selected instances from each dataset, with three people reviewing each instance. All three reviewers agreed on the label for 91% of the generated data. For 94% of the data, at least two reviewers agreed on the label.

### A.5 HUMAN EVALUATION ON BIRD ESTIMATED PROBABILITIES

We first explain our choice of relative evaluation over asking human annotators to provide absolute probabilities from a psychological perspective. TVERSKY & KAHNEMAN (1974) shows that humans use heuristic principles to make probability judgments, which often suffer from systematic and predictable errors due to cognitive biases. Kahneman (2003) further discusses how human judgment in probabilistic situations is generally better in relative than absolute terms due to cognitive biases and limitations in processing complex information. We conducted a pilot study with human

---

[8]We use GPT-4-0613 when utilizing GPT-4 for annotations.

---

**Algorithm 1:** Training single probabilities to estimate $\mathbb{P}(O_i|f)$ .

---

**input** : Training set of complete information $\{f\}, f \in \mathcal{F}$, outcome $O_i, i = 1, 2$, LLM $M$, learning rate $\eta$, number of epochs $E$, batch size $B$.

**output** : Trained probabilities $\mathbb{P}(O_i|f_{ja}), f_{ja} \in \mathcal{F}_j, j = 1, ..., N, a = 1, ..., \mathrm{card}(\mathcal{F}_j), i = 1, 2.$

  # Get verbalized probability scores from the LLM.

1 **for** $f^{'} \in \{f\}$ **do**

2    $\mathbb{P}_{\mathrm{LLM}}(O_1|f^{'}), \mathbb{P}_{\mathrm{LLM}}(O_2|f^{'}) \leftarrow M(f^{'}, O_1, O_2)$

3 **end**

  # Initialize probabilities and calculate target for the margin
  ranking loss.

4 **for** $j = 1$ **to** $N$ **do**

5    **for** $a = 1$ **to** $\mathrm{card}(\mathcal{F}_j)$ **do**

6       $\mathbb{P}_{\mathrm{init}}(O_1|f_{ja}) = \begin{cases} 75\% & f_{ja} \text{ supports outcome 1} \\ 50\% & f_{ja} \text{ is neutral} \\ 25\% & f_{ja} \text{ supports opposite outcome 2} \end{cases}$

7       $y_{\mathrm{target}}(O_1|f_{ja}) = \begin{cases} 1 & \mathbb{P}_{\mathrm{init}}(O_1|f_{ja}) > 50\% \\ 0 & \mathbb{P}_{\mathrm{init}}(O_1|f_{ja}) = 50\% \\ -1 & \mathbb{P}_{\mathrm{init}}(O_1|f_{ja}) < 50\% \end{cases}$

8    **end**

9 **end**

  # Training.

10 **for** $epoch = 1$ **to** $E$ **do**

11    **for** *each batch of B samples from the training set* **do**

12       **for** *$b \in 1, ..., B$(in parallel)* **do**

13          $\mathbb{P}_{\mathrm{estimated}}(O_1|f^b) = \frac{\prod_{j=1}^N \mathbb{P}(O_1|f_j^b)}{\prod_{j=1}^N \mathbb{P}(O_1|f_j^b) + \prod_{j=1}^N (1 - \mathbb{P}(O_1|f_j^b))}$  # Forward propagation.

14          $\ell_{\mathrm{MSE}}^b = (\mathbb{P}_{\mathrm{LLM}}(O_1|f^b) - \mathbb{P}_{\mathrm{estimated}}(O_1|f^b))^2$ # Compute MSE loss for each sample, for $f_{ja} \in \mathcal{F}_j, f_{ja} \neq f_j^b, \frac{\partial \ell_{\mathrm{MSE}}^b}{\partial \mathbb{P}(O_1|f_{ja})} = 0, j = 1, ..., N$.

15       **end**

16       $\ell_{\mathrm{MSE}} = \frac{1}{B} \sum_{b=1}^B \ell_{\mathrm{MSE}}^b$ # Compute MSE loss.

17       **for** $j = 1$ **to** $N$ **do**

18          **for** $a = 1$ **to** $\mathrm{card}(\mathcal{F}_j)$ **do**

19             $\mathbb{P}_{\mathrm{trained}}(O_1|f_{ja}) = \sum_{f^* \in \mathcal{F}^*} \mathbb{P}_{\mathrm{estimated}}(O_1|f^* \cup f_{ja}) \mathbb{P}(f^*|f_{ja}) = \frac{1}{\mathrm{card}(\mathcal{F}^*)} \sum_{f^* \in \mathcal{F}^*} \mathbb{P}_{\mathrm{estimated}}(O_1|f^* \cup f_{ja})$

20             $\ell_{\mathrm{MR}}^{ja} = \max(0, -y_{\mathrm{target}}(O_1|f_{ja})(\mathbb{P}_{\mathrm{trained}}(O_1|f_{ja}) - 0.5) + \epsilon)$

21          **end**

22       **end**

23       $\ell_{\mathrm{MR}} = \frac{1}{N} \sum_{j=1}^N \frac{1}{\mathrm{card}(\mathcal{F}_j)} \sum_{a=1}^{\mathrm{card}(\mathcal{F}_j)} \ell_{\mathrm{MR}}^{ja}$ # Compute MR loss.

24       $\mathcal{L} = \ell_{\mathrm{MSE}} + \alpha * \ell_{\mathrm{MR}}$ # Compute final loss.

25       **for** $j = 1$ **to** $N$ **do**

26          **for** $a = 1$ **to** $\mathrm{card}(\mathcal{F}_j)$ **do**

27             $\mathbb{P}(O_1|f_{ja}) = \mathbb{P}(O_1|f_{ja}) - \eta \frac{\partial \mathcal{L}}{\partial \mathbb{P}(O_1|f_{ja})}$  # Update parameters.

28             $\mathbb{P}(O_2|f_{ja}) = 1 - \mathbb{P}(O_1|f_{ja})$

29          **end**

30       **end**

31    **end**

32 **end**

---

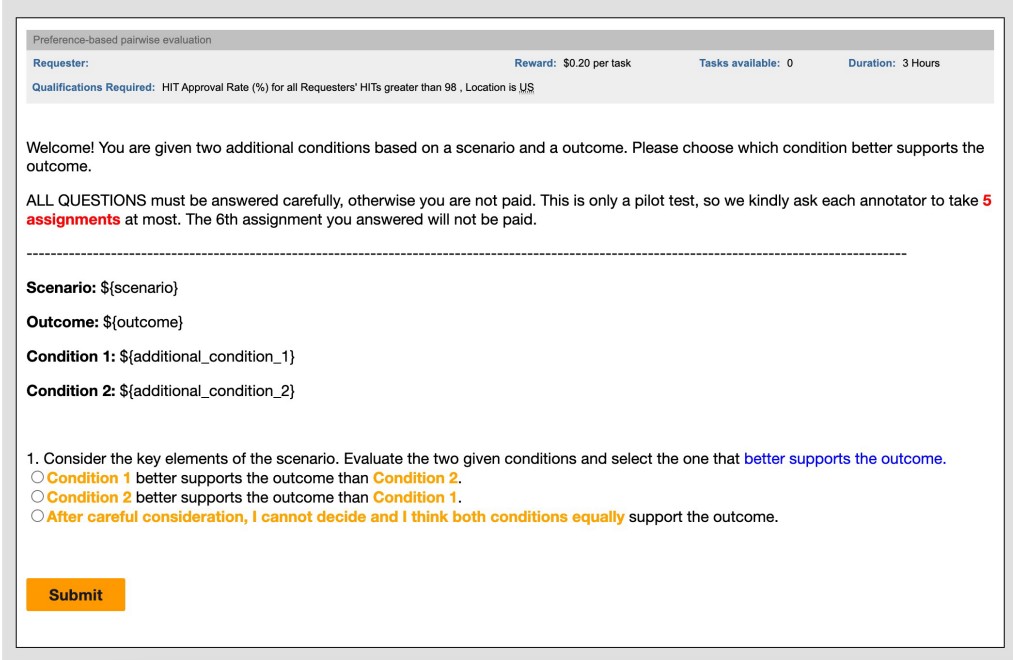

Figure 3: The interface for human evaluation on preference-based pairwise evaluation of the estimated probabilities.

annotators on MTurk, asking them to provide absolute probabilities for each additional condition separately. However, we were unable to reach a consensus among the 7 annotators.

We then introduce the human annotation process for pairwise comparison. We collect human gold labels with rigorous quality control for the preference-based pairwise evaluation of the estimated probabilities. We restrict each annotator to take 5 samples at most and there are 386 distinct annotators. We only allow participants who have 98% or more HITs acceptance rate and are located in the US. We also require annotators to spend at least 1 minute for each instance (the hourly salary is $15). We show the experiment interfaces in Appendix Fig. 3. We first carry out the human annotation process five times on 450 randomly sampled instances from COMMON2SENSE when the condition can be mapped to at least one factor and therefore BIRD does not produce "unknown" during our experiments, and both conditions support one target outcome. If the 5 annotations have a less than 80% agreement, i.e., less than 4 out of the 5 annotators agree on the same condition label, we invite 2 additional annotators to reassess this instance. If the agreement of the 7 annotators is more than 70%, i.e., at least 5 out of the 7 annotators agree on the same condition label, we accept the instance. Other than the previously accepted instances, we also accept an instance as a neutral label if all three labels have been chosen during the annotations and the agreement for the neutral label is more than or equal 40% and the agreement for any of the remaining two labels is less than or equal 40%. We get 350 instances in total, where 12.3% of the instances have a neutral label where condition 1 and condition 2 equally support the outcome, 44.0% of the instances have a label towards condition 1 where condition 1 supports the corresponding outcome better than condition 2, 43.7% of the instances have a label towards condition 2 where condition 2 supports the corresponding outcome better than condition 1.

We now introduce how precision and recall are calculated. Given a testing instance, assuming the gold outcome is $O^*$ and the opposite outcome is $O^\neg$, there will be 3 model output labels: 1) $P(O^*|C1) > P(O^*|C2)$, i.e., the model thinks C1 better support the gold outcome; 2) $P(O^*|C2) > P(O^*|C1)$, i.e., the model thinks C2 better support the gold outcome; 3) $P(O^*|C2) = P(O^*|C1)$, i.e., the model thinks both $C_1$ and $C_2$ equally support the gold outcome. We want to check if the model's label is the same as the human annotator's label. Therefore, this can further be seen as a 3-label classification problem.

## A.6 BASELINE DESCRIPTION

We adopt the following baselines to evaluate our BIRD framework.

- **Vanilla Verbalization with Self-Consistency (Vanilla).** (Wang et al., 2023) Given the scenario, an additional condition, and two corresponding outcomes, the model is asked to directly verbalize an estimated probability for each outcome. We show the Prompt in Appendix Fig. 11. The above process is carried out independently for each of the two additional conditions. The additional conditions with higher output probabilities for the gold outcome are the predicted answers. We use temperature = 0.7 to generate 3 responses and take the majority vote answer as the final answer.

- **Logit-based Probability.** We directly use the output token probability for the decision. We use greedy decoding here to set the temperature = 0.

- **Chain-of-Thought Verbalization with Self-consistency (CoT).** (Wei et al., 2022) We ask for a reasoning process to think step by step before verbalizing an estimated probability. The remaining process is the same as vanilla verbalization with SC. We show the Prompt in Appendix Fig. 12.

- **Explicit Comparison (EC).** Instead of evaluating each additional condition separately, both conditions are presented simultaneously to the model. Given the scenario and the gold outcome, the model is asked to select which additional condition better supports the outcome without outputting any probabilities. Note that the task is less challenging for this particular baseline than the others due to the explicit formatting of inference in this baseline. As a result, performances under the EC setting are not directly comparable with our model and other baselines' performances since they only see one condition at a time and cannot perform argmax operations. We show the Prompt in Appendix Fig. 13.

- **1/2 Assumption.** Instead of utilizing the learning algorithm mentioned in §3.3 to estimate $\mathbb{P}(O_i|f)$. We adopt a neutral assumption. An outcome is determined based on the factors only if, except the values of the factors that are neutral, the values of all remaining factors uniformly support that particular outcome. Conversely, if the values of the factors indicate different outcomes and there is no prior information, we cannot conclusively determine the outcome.

$$\mathbb{P}(O_i|f) = \left\{ \begin{array}{ll} 1 & \mathbb{P}(O_i|f_j) \geq \frac{1}{2}, \forall j \text{ except } \mathbb{P}(O_i|f_j) = \frac{1}{2}, j = 1, ..., N \\ \frac{1}{2} & \text{otherwise} \end{array} \right. \tag{12}$$

The intuition behind this assumption is that as we focus solely on the present context, if the current information supports both outcomes and no additional information is available to finalize the decision, it becomes impossible to predict which outcome is more likely to occur.

- **1/N Assumption.** Instead of utilizing the learning algorithm mentioned in §3.3 to estimate $\mathbb{P}(O_i|f)$, we adopt another commonly used neutral assumption. The outcome $O_i$'s probability is determined by the aggregate count of factors whose values support outcome $O_i$.

$$\mathbb{P}(O_i|f) = \frac{\sum_{j=1}^{N} \mathbb{1}\{\mathbb{P}(O_i|f_j) > \frac{1}{2}\}}{N} \tag{13}$$

- **Fixed Initial Prob.** Instead of training with the algorithm to get an estimated prob for each $\mathbb{P}(O_i|f_j)$ in §3.3, we utilize the initial probability of Eq.5 to directly estimate $\mathbb{P}(O_i|f)$ with Eq.4.

## A.7 ANALYSIS OF DIRECT DECISION MAKING

We first conduct error analysis for Llama-2-70b-instruct on direct decision making where BIRD gets the wrong outcomes while CoT makes the correct decisions. We detail the type of errors with its frequency percentage in Appendix Table 4. We observe that our proposed method outperforms the chain-of-thought baseline with Llama-3.1 rather than using Llama-2. This is partially because Llama-3.1 generates more diverse and accurate factors, and we present an example in Appendix Table 5.

| Error Type | Frequency Percentage |
|---|---|
| The generated factors and their values are not comprehensive enough to cover all elements of the conditions. The selected values only imply partial information in the additional condition. | 56.7% |
| The additional condition is mapped to incorrect values of the factors that it does not necessarily imply. | 30% |
| The calculation of the final probability is inaccurate. | 13.3% |

Table 4: Error analysis for Llama-2-70b-instruct for direct decision making.

**Scenario**: the ease of carrying a cup is being compared between one person and six people.
**Outcome 1**: It would be easier for one person to carry the cup, than it would be for six people to carry the cup.
**Outcome 2**: It would be easier for six people to carry the cup, than it would be for one person to carry the cup.

**Llama-2-70b-Instruct Generated Factors**

**Weight and Size of the Cup**:
The cup is light and small
The cup is heavy and large
The cup is of medium weight and size
**Material of the Cup**:
The cup is fragile
The cup is sturdy
The cup is made of moderately durable material
**Strength of the People**:
The person is able-bodied and strong enough to carry the cup
The combined strength of six people is greater than that of one person
**Experience of the Person or People Involved**:
The person has more experience in carrying objects
The six people have more experience in lifting objects
**Formation of the People**:
The six people carry the cup in a specific formation designed to distribute the weight more evenly
The six people do not carry the cup in a specific formation

**Llama-3.1-70b-Instruct Generated Factors**

**Cup Size and Weight**:
The cup is small and light
The cup is large and heavy
The cup is small and heavy
The cup is large and light
**Cup Material**:
The cup is fragile
The cup is sturdy
The cup is constructed from a material of moderate durability
**Cup Design and Features**:
The cup has a comfortable grip and balanced ergonomics
The cup has an uncomfortable grip and unbalanced ergonomics
The cup has a neutral grip and average ergonomics
**Carrier's Physical Abilities**:
The one carrier has more strength and endurance than the six carries
The combined strength of the six carries is stronger than the one carrier
**Group Dynamics and Coordination**:
The group is well-coordinated and works together seamlessly
The group is disorganized and has difficulty working together
The group has average coordination and works together moderately well
**Physical Demands of the Task**:
The task requires exceptional strength and endurance
The task requires minimal strength and endurance
The task requires moderate strength and endurance
**The Size and Shape of the Carrying Area**:
The carrying area is narrow and confined
The carrying area is wide and open

Table 5: An example where Llama-3.1 generates better factors than Llama-2.

We then give insights into LLM direct performance. We present an example from the COMMON2SENSE dataset in Appendix Table 6. where the LLM accurately infers the correct outcome based predominantly on inductive biases, ignoring the other primary information provided in the condition, i.e., the peppers are used in large quantities, which will be captured in our framework.

---

**Example**

Scenario: The spiciness of a hot sauce is being discussed in relation to the choice between using habanero peppers or jalapeno peppers.
Condition: The hot sauce recipe uses a significant amount of habanero peppers relative to the quantity of other ingredients.
Outcome 1: A hot sauce is going to be hotter if it uses habanero peppers rather than jalapeno.
Outcome 2: A hot sauce is going to be hotter if it uses jalapeno peppers rather than habanero.

**Llama-2 Generated Deicision Making Process**

Habanero peppers are generally much hotter than jalapeno peppers. This is measured on the Scoville scale, which is a measure of the pungency (spiciness or heat) of chili peppers and other spicy foods. So the condition better supports Outcome 1: A hot sauce is going to be hotter if it uses habanero peppers rather than jalapeno.

---

Table 6: An example where the LLM correctly makes the decision but with inadequate emphasis on the context.

## A.8   ABLATION STUDY TOWARDS ABDUCTIVE FACTOR GENERATION

We compare the final decision-making accuracy based on two different factor generation methods and present the results in Appendix Table 7. We see that our proposed method performs much better than the direct generation method, indicating the effectiveness of utilizing the model's inductive bias inherited from training data.

| Dataset | Method | $\text{BIRD}_{llama2}$ | %Unknown | $\text{BIRD}_{llama3.1}$ | %Unknown |
|---------|--------|------------------------|----------|--------------------------|----------|
| TODAY   | Direct | 70.9 | 74.1 | 68.9 | 46.3 |
|         | Abduction | 73.9 | 54.0 | 74.3 | 34.0 |
| PLASMA  | Direct | 68.6 | 40.7 | 69.0 | 24.1 |
|         | Abduction | 74.0 | 31.4 | 73.0 | 19.0 |
| C2S     | Direct | 84.2 | 54.9 | 89.9 | 29.0 |
|         | Abduction | 89.0 | 34.6 | 92.3 | 27.1 |

Table 7: An ablation study towards different factor generation methods. Direct refers to the direct factor generation method. Abduction refers to our abductive factor generation.

## A.9   ABLATION STUDY TOWARDS THE EFFECT OF THE TWO LOSS FUNCTIONS IN THE LEARNING ALGORITHM

We conduct an ablation study on the loss function in Eq.7 for decision making, as detailed in Appendix Table 8, and demonstrate that both components of the loss function are crucial for final performance.

| Dataset | w/o MR Loss | w/o MSE Loss | BIRD |
|---------|-------------|--------------|------|
| TODAY   | 65.5 | 65.2 | 74.3 |
| PLASMA  | 60.0 | 65.7 | 73.0 |
| C2S     | 77.2 | 86.8 | 92.3 |

Table 8: An ablation study towards the effect of the two loss functions in Eq.7 for decision making with Llama-3.1-70b-Instruct.

### A.10 EXPERIMENT SETTINGS FOR USING ESTIMATED PROBABILITIES AS ADDITIONAL TRAINING DATA

The T5-large model and PatternTime (Zhou et al., 2021) are specialized temporal reasoning models, as outlined in Feng et al. (2023). Both models were specifically fine-tuned on temporal reasoning datasets using the original temporal training data from Feng et al. (2023). In contrast, the additional training data from COMMON2SENSE in §4.3 focuses on commonsense reasoning.

### A.11 YES-OR-NO FOLLOWUP QUESTION GENERATION

We experiment to see if BIRD can further facilitate decision making in a controllable way, where we use BIRD to generate a yes-or-no follow-up question about an unobserved factor in the context. Specifically, given a scenario and an additional condition, we first identify which factors and their corresponding values are implied by the condition, we then select a factor that is not mentioned and is most important for the decision in the remaining unmentioned factors, as the target question area. We adopt the factor value that better aligns with the outcome direction suggested by the current implied factor values and has the largest condition probability. We then rewrite the statement of the factor value into its corresponding question. Simultaneously, we ask Llama2-70b-instruct to directly generate a yes-or-no follow-up question. The prompt is in Appendix Fig. 15.

We use MTurk and ask annotators to provide pairwise preference between the BIRD generated question and LLM directly generated question. We adopt the same Mturk setting as mentioned in §A.5. we carry out the human annotation process three times, using the majority vote to find the preferred generated question. We show the experiment interfaces and an example in Appendix Fig. 4. Appendix Table 9 shows that the follow-up questions generated by BIRD are more often preferred.

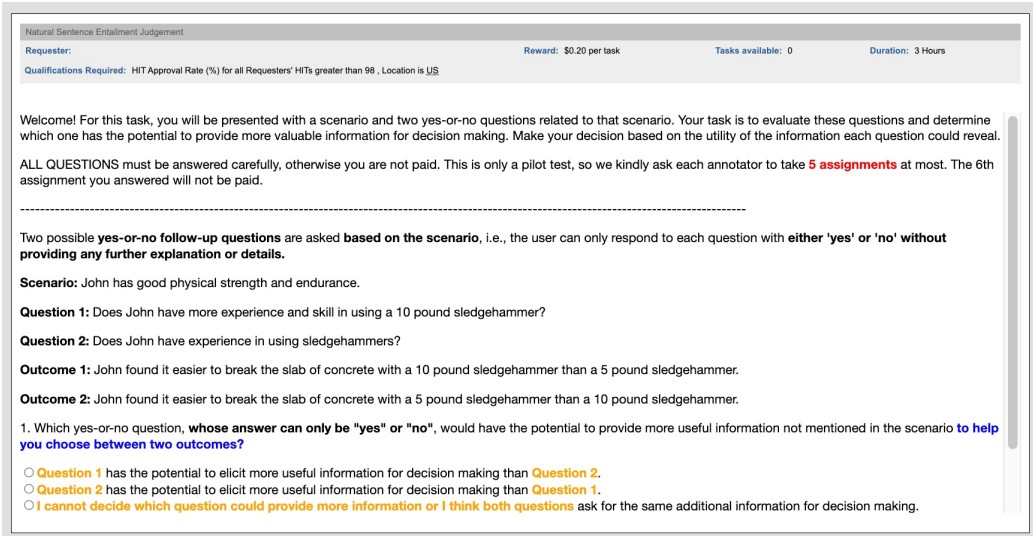

Figure 4: The interface for human evaluation on preference-based pairwise evaluation of the generated questions. Question 1 is generated by BIRD and question 2 is generated by LLM directly. The three annotators all prefer question 1.

| Dataset | BIRD Win Direct | Tie | BIRD Lose Direct |
|---|---|---|---|
| COMMON2SENSE | 132 | 36 | 82 |

Table 9: Preference judgments for yes-or-no follow-up questions generated by BIRD versus questions directly generated by LLMs. We randomly sample 250 instances from COMMON2SENSE.

### A.12 PROMPT

EXAMPLE **ADDITIONAL SENTENCE SAMPLING** PROMPT

**SYSTEM**
You are given a scenario and an accompanying hypothesis. Generate 5 sentences covering different conditions that would add objective information relevant to the hypothesis such that the hypothesis is more likely to hold true. The information should not definitively imply the hypothesis. You must follow the below structure to just generate sentences with no explanations.

**USER**
Scenario: You want to move around with your cell phone when it is being charged.
Hypothesis: You can move around more freely with your cell phone if it is being charged with a one-foot cord rather than a six-foot cord.

**ASSISTANT**
# The cell phone is being charged with a portable power bank located in your pocket, allowing you to move around without being tethered to a fixed outlet.
# The user is working in a compact space where longer cords could easily snag on furniture or equipment, thus a one-foot cord could minimize this risk.
# The phone is needed for tasks that require frequent handling and close proximity to the user, making a shorter cord more practical to avoid excessive dangling.
# The charging setup includes a small desktop charger that keeps the phone elevated and stable, limiting the practicality of a longer cord.
# The user is in a busy environment like a kitchen or workshop, where shorter cords can reduce the hazard of tripping or catching on moving objects.

Figure 5: Example **Additional Sentence Sampling** Prompt

---

**EXAMPLE FACTOR SUMMARIZATION FROM ADDITIONAL SENTENCES PROMPT**

**SYSTEM**
From the given sentences for each outcome, identify and list
distinct and concrete factors, ensuring each is broad yet specific
and focuses on a unique aspect.
Your response should strictly adhere to the JSON format provided,
without additional explanations.
For example: {"distinct factor" <ENSURE each factor focuses on a
unique aspect>: "factor values" <Each factor MUST cover at least
one condition to support the Statement and one condition to support
the Opposite statement.>}
1. Ensure that each factor's value MUST directly reference
specific elements mentioned in the statements, avoiding vague terms
like 'the object'.
2. Ensure the factor values are not too concrete.
3. Do not only mention the common situations.

**USER**
Scenario: You are charging your cell phone and wish to move around
with your cell phone.
Outcome 1: You can move around more freely with your cell phone
if it is being charged with a six feet cord rather than a one feet
cord.
Sentences:
#1 A longer cord provides more flexibility and allows for a greater
range of movement while using the cell phone. This is because the
additional length of the six-foot cord gives the user a larger
radius of movement, enabling them to comfortably use their phone
while it is charging without feeling restricted or confined to a
specific location.
Outcome 2: You can move around more freely with your cell phone
if it is being charged with a one-foot cord rather than a six-foot
cord.
Sentences:
#2 If the cell phone is plugged into a portable power bank or a
USB port on a computer, a one-foot cord provides greater mobility
because it is shorter and less likely to get tangled or caught on
objects while moving.
#3 If the cell phone is constantly being used while charging and
the user prefers to keep the phone close to the charger at all
times, a one-foot cord allows for easier mobility and reduces the
risk of tripping over a longer cord."

**ASSISTANT**
{ "The cell phone's charging method":
["The charger is portable", "The charger is unmovable"],
"The user's movement range":
["The user stays very close to the charger", "The user has a large
radius of movement"],
"The location of the phone charger":
["The user leaves the charger somewhere", "The user carries the
charger"]}

Figure 6: Example **Factor Summarization from Additional Sentences** Prompt

---

**EXAMPLE PROMPT FOR DECIDING THE VALUE BETTER SUPPORTS WHICH OUTCOME**

**SYSTEM**
Decide whether each of the given values supports Statement 1 or
Statement 2. The statement and opposite statement are provided,
along with a list of values. Determine which statement each value
supports. You must follow the below structure to just generate
json with an explanation.
For example:
Explanation: <explanations for how each value supports which
statement>
List: { "value 1":<output only Statement 1, Statement 2, Both, or
Neither>, "value 2":<output only Statement 1, Statement 2, Both, or
Neither>}
**USER**
Here is a scenario: ... I'd like you to determine which statement
each value supports.
Values: ...
* Statement 1: ...
* Statement 2: ...
Please analyze each value and determine which statement it
supports.
**ASSISTANT**
Explanation: [step-by-step analysis here]
List: { "value":"Statement 1" }

Figure 7: Example Prompt for **Deciding the Value Better Supports Which Outcome**

---

**EXAMPLE PROMPT FOR DECIDING IF THE SCENARIO WITH THE CONDITION IMPLIES THE VALUE OF THE FACTOR**

**SYSTEM**
Given a scenario, read a sentence and determine whether a value
from a given list is implied by the sentence. You must be strict
on considering a value as implied and return one value or "None"
with an explanation.
Here are rules that you must follow:
1. A value is implied only if the sentence implies it in most
situations, meaning the sentence provides clear, direct information
relevant to the value.
2. Do not select a value based on indirect associations, context,
or related hints unless they strongly imply the value in most
situations.
3. Do not assume causality or infer a connection if it is not
clearly stated in the sentence.
4. If the sentence implies more than one value, choose the most
implied one. If two values are both implied, choose the one that
is positive (non-negative).
5. If the sentence does not imply any value directly in most
situations, say "None".
**USER**
Here is a scenario: ...
Sentence: ...
List: "factor": [ "value 1", ... ]
**ASSISTANT**
{"explanation": ...,
"answer": a value/None}

Figure 8: Example Prompt for **Deciding If the Scenario with the Condition Implies the Value of the Factor**

---

EXAMPLE PROMPT FOR **REEVALUATING IMPLIED VALUE**

**SYSTEM**
Based on a scenario, delete value in the list if the key does not
imply value. You should output an explanation and then a json.
You must generate double quotes.
Here are rules that you must follow:
- Ensure that retained values align with the key's implications,
do not assume causality or infer a connection if the value is not
clearly stated in the key. Avoid over-generalizations and ensure
logical consistency in your decisions.
- Be lenient with deletions. Only remove a value if the key clearly
does not imply the value.
- Provide a clear explanation for each deletion or retention.

**USER**
Scenario:  The government is planning the locations for building
charging stations.
{"The location is on a busy highway with no existing charging
stations.":
["No nearby charging stations",
"Location is on a major travel route, serving long-distance EV
travelers",
"Nearby amenities like restaurants, shops, and rest areas for users
while charging"]}

**ASSISTANT**
Explanation:  "No nearby charging stations" was kept because the
key explicitly mentions the absence of existing charging stations
in the area.  Similarly, "Location is on a major travel route,
serving long-distance EV travelers" was retained because being on a
busy highway strongly implies that it serves as a key travel route
for such travelers.  However, "Nearby amenities like restaurants,
shops, and rest areas for users while charging" was removed since
the key does not directly mention or suggest the presence of these
amenities.
List:  {"The location is on a busy highway with no existing
charging stations.":
["No nearby charging stations",
"Location is on a major travel route, serving long-distance EV
travelers"]}

Figure 9: Example Prompt for **Reevaluating Implied Value**. Explanation is optional.

---

**EXAMPLE PROMPT FOR OUTPUTTING A VERBALIZED PROBABILITY SCORE**

**SYSTEM**
```
As an AI assistant, your role is to respond accurately to user
queries.  While answering think step-by-step and justify your
answer.  Analyze the given scenario and condition to determine the
likelihood of the outcomes.  Use only the information provided,
without relying on prior knowledge.  Assess the probability using
the specified terms:  'very likely', 'likely', 'somewhat likely',
'neutral', 'somewhat unlikely', 'unlikely', 'very unlikely'.  Ensure
that your assessments are complementary:  if one outcome is deemed
'likely', the other must be 'unlikely', and so on.  You should first
give your reasons and then format your final answer.
```
**USER**
```
Scenario:  You want to move around with your cell phone when it is
being charged.
Condition:  The charger is portable.  The user stays very close to
the charger.  The user carries the charger.
Outcome 1:  You can move around more freely with your cell phone
with a one-foot cord rather than a six-foot cord.
Outcome 2:  You can move around more freely with your cell phone
with a six-foot cord rather than a one-foot cord.
```
**ASSISTANT**
```
Given that the user is carrying a portable charger, a shorter cord
like one foot would indeed be far more manageable, making it easier
for the user to move around freely.
Given the same conditions, a longer cord like six feet might become
an impediment, making it more challenging for the user who is
carrying the charger to move around freely due to the possibility
of tangling or managing the extra length.
Therefore, Outcome 1:  Likely
Outcome 2:  Unlikely
```

Figure 10: Example Prompt for **Outputting a Verbalized Probability Score**

---

**EXAMPLE PROMPT FOR DIRECT PROBABILITY VERBALIZATION BASELINE**

**SYSTEM**
```
You are an AI assistant that helps people make decisions.  The user
will give you a question.  Choose which outcome the scenario and
its corresponding additional condition better support.  Provide
your confidence level as a percentage to indicate how certain you
are about your answer.
```
**USER**
```
Scenario:  ...
Additional Condition:  ...
Outcome 1:  ...
Outcome 2:  ...
```
**ASSISTANT**
```
Answer and Confidence (0-100):  Outcome i, X
```

Figure 11: Example Prompt for **Direct Probability Verbalization Baseline**

---

EXAMPLE PROMPT FOR **DIRECT PROBABILITY VERBALIZATION WITH COT BASELINE**

**SYSTEM**
```
You are an AI assistant that helps people make decisions.  The user
will give you a question.  Your task is to answer as faithfully
as you can.  While answering, think step-by-step and justify
your answer.  You are asked to provide a probability for each
outcome being correct (0% to 100%) based on the scenario and its
corresponding additional condition.  Ensure that P(outcome 1) +
P(outcome 2) = 1 .
```
**USER**
```
Scenario:  ...
Additional Condition:  ...
Outcome 1:  ...
Outcome 2:  ...
```
**ASSISTANT**
```
Explanation:  [step-by-step analysis here]
Confidence:  P(outcome 1) = X%, P(outcome 2) = Y%.
```

Figure 12: Example Prompt for **Direct Probability Verbalization with CoT Baseline**

---

EXAMPLE PROMPT FOR **EXPLICIT COMPARISON BASELINE**

**SYSTEM**
```
You are an AI assistant that helps people make decisions.  The user
will give you a question.  Your task is to answer as faithfully
as you can.  While answering, think step-by-step and justify your
answer.  You are asked to decide which condition makes the outcome
more likely to be true given the scenario.
```
**USER**
```
Scenario:  ...
Outcome:  ...
Condition 1:  ...
Condition 2:  ...
```
**ASSISTANT**
```
Explanation:  [step-by-step analysis here]
Answer:  Condition i/Both.
```

Figure 13: Example Prompt for **Explicit Comparison Baseline**

---

**EXAMPLE COT FOR DECISION MAKING PROMPT**

**SYSTEM**
You are a helpful assistant. While answering think step-by-step and justify your answer. Given a scenario and an additional condition, decide which of the two outcomes is supported.
**USER**
Scenario: The government is planning the locations for building charging stations.
Condition: The location is on a busy highway with no existing charging stations.
Outcome 1: The government should build a charging station here.
Outcome 2: The government should not build a charging station here.
**ASSISTANT**
The scenario presents a busy highway with no existing charging stations, and the government is planning where to build new ones. Given the high traffic volume and the absence of charging infrastructure, building a station here would fill a critical gap, support electric vehicle drivers, and encourage the transition to sustainable transportation.
Based on the available information, Outcome 1: The government should build a charging station here is more supported.

Figure 14: Example **COT for Decision Making** Prompt

---

**EXAMPLE DIRECT YES-OR-NO FOLLOW-UP QUESTION GENERATION PROMPT**

**SYSTEM**
You are a helpful assistant. While answering think step-by-step and justify your answer. Ask a follow-up yes or no question so that you can better decide which outcome is better supported by the scenario and the condition. You can only ask for a specific situation.
**USER**
Scenario: You want to move around with your cell phone when it is being charged.
Condition: The charger is portable and the user carries the charger.
Outcome 1: You can move around more freely with your cell phone with a one-foot cord rather than a six feet cord.
Outcome 2: You can move around more freely with your cell phone with a six-foot cord rather than a one feet cord.
**ASSISTANT**
In this scenario, the primary factor to consider is cord length, in most situations, a greater cord length allows more movement freedom, as you can be further from the outlet and still keep your device charged. However, considering that user carries the charger, it might bring about the confusion between the two outcomes.
Thus, to further ascertain which outcome is supported better, I'd ask the following question:
Does the portable charger need to be plugged into an outlet to charge the phone?

Figure 15: Example **Direct Yes-or-no Follow-up Question Generation** Prompt

---

