# OpenReview forum: "BIRD: A Trustworthy Bayesian Inference Framework for Large Language Models"
_ICLR.cc/2025/Conference — ICLR 2025 Oral_

### Official Review · Reviewer_VqFc · 2024-10-21

**Soundness:** 4
**Presentation:** 3
**Contribution:** 3
**Rating:** 8
**Confidence:** 3

**Summary:**

This paper proposes a BIRD framework that enables more reliable probability estimation based on incomplete instances from LLMs. It leverages factors generated by LLMs to make a Bayesian network and estimates more precise probabilities. BIRD with optimized prob presents a better performance in several experiments than existing methods. BIRD also demonstrates improved reasoning performance across various datasets, particularly excelling in more complex scenarios.

**Strengths:**

1) The proposed framework and optimization settings, which account for weak ordering and non-interaction in the set of factors, are valid.
2) The authors' assumptions about enhancing the validity of the proposed Bayesian Inference Framework are reasonable.
3) Including unobserved factors in the calculations strengthens the effectiveness and utility of the authors' proposed method.
4) Besides precise probability estimation, the presence of follow-up generation and cross-domain experiments demonstrates the proposed method's reliability and practicality.

**Weaknesses:**

1) In Line 147 and Line 155, the explanation of the outcomes is separated. It seems possible to consolidate these explanations into one part. Additionally, the explanation of 𝐹 in the same paragraph could be made more concise. The space saved from these revisions could then be used to more clearly elaborate on the connection between Equation (5) and Equation (6) and the content in Algorithm 1.

2) Ablation settings could be expanded to cover more aspects. For example, experiments could be conducted by excluding the MR loss from the unified loss function or by assuming that in an unobserved factor, each value does not have an equal probability of being selected.

**Questions:**

1) Page 1: LLMs’s → LLMs’
2) The LaTeX expression for P in the introduction differs from its expression in the later sections.
3) Where is the example of the factor, "The color of the vehicle," mentioned by the authors in Line 228, depicted in Figure 2? I couldn't find it in the figure as described in the text.
4) Appendix A.2: Week →Weak
5) This may be a minor issue, but there could be confusion between Line 318 'S + U₃ implies d₁ and f₂' and f, which is a possible instance of complete information due to the notation.
6) An additional question is, have the authors considered any experiments involving more than two outcomes (i.e., beyond binary classification)?

---

> ### Author Response · Authors · 2024-11-20
>
> We sincerely thank reviewer VqFc for the valuable comments. Here we answer all your questions.
>
> **W1.Paper writing**
>
> We have addressed the minor writing issues in the updated version. While we agree that the writing for the general setting could be made more concise and that the algorithm can be explained in greater detail (many details have been added in Appendix A.3), we plan to address these aspects more comprehensively in the final version as structural changes to the writing require a relatively large amount of time.
>
> We are displaying factors related to the decision making in Figure 2, and “the color of the vehicle” is not, so it does not appear in the figure.
>
>
> **W2. Ablation studies**
>
> We conduct a preliminary ablation study on the loss function for decision making, as detailed below, and demonstrate that both components of the loss function are crucial for the model's performance. We will include the complete ablation study in the final version.
>
> | Llama-3.1-70b-Instruct | w/o MR Loss | w/o MSE Loss | BIRD  |
> |-------------------------|-------------|--------------|-------|
> | TODAY                  | 65.5        | 65.2         | 74.3  |
> | Plasma                 | 60.0        | 65.7         | 73.0  |
> | Common2sense           | 77.2        | 86.8         | 92.3  |
>
> We believe that assuming each value in an unobserved factor has an equal probability of being selected is an unbiased and neutral assumption, given the lack of information about which value might be chosen. However, we agree with the reviewer that further user interaction could be beneficial in determining if there are specific preferences regarding the possible values.
>
>
> **Q6. Decision-making involving more than two outcomes**
>
> We agree with the reviewer that testing decision-making scenarios with more than two outcomes is an important next step. BIRD can handle multiple outcome situations by hierarchically iterating pairs of outcomes within the decision-making framework. Additionally, future work could focus on deriving new guarantees, similar to Equation 3, for multiple outcomes, which could lead to the formulation of a new constrained optimization problem.

---

> > ### Comment · Reviewer_VqFc · 2024-11-28
> >
> > Thank you for your response. I believe the paper received a score that matches its quality.

---

### Official Review · Reviewer_ZTvP · 2024-10-28

**Soundness:** 3
**Presentation:** 4
**Contribution:** 4
**Rating:** 8
**Confidence:** 3

**Summary:**

This paper presents a method allowing LLMs to reason more soundly about complex problems with uncertainty, by combining LLM introspection and verbal estimates of uncertainty into Bayesian reasoning about outcomes. In many ways it resembles chain-of-thought, in that the LLM is being asked to provide high level reasoning about an uncertain scenario, and is required to brainstorm to solve the problem. The incorporation of a Bayes net sets the method apart from prior work, and experiments show that this leads to much better estimates in pairwise evaluation of propositions, but less compelling evidence for the work when evaluated on decision making benchmarks.

**Strengths:**

This paper presented a very novel idea, a detailed method for achieving this, and a convincing evaluation of the utility of the technique. Specific strengths include:
 - mechanism for prompting a LLM to extract factors & potential outcomes of a complex decision problem
 - means to convert the factors into a binary Bayes network
 - incorporation of model uncertainty estimates into the parameterisation of the Bayes network
 - evaluation against human judgements of probability
 - strong baselines

**Weaknesses:**

The paper was quite dense to start, and took several pages to make the application scenario clear. On my first pass it wasn't apparent to me what elements of the data were part of the problem definition, what parts were computed by the LLM, and what was computed by inference in a Bayes net. Fig 1 is reasonably easy to follow, but Fig 2 is less clear - why have the conditions changed between Fig 1 and Fig2, and what parts of the central box are given as part of the data versus inferred from LLM interactions? As I understand it, the whole of the inner box is auto-generated, as well as the edges linking the given conditions with coarse probability estimates.

The paper also takes a long and detailed dive into Bayes net formality, which is more than needed given the setting is relatively simple and could be communicated effectively with more of a focus on the running example (Fig 2 could be unpacked). As it stands, the paper takes several pages to get going, but ends up deferring many critical details to the Appendix. The method for prompting the LLM is of fundamental importance to the technique, for instance, but none of this is in the main paper.

**Questions:**

There several magic numbers in 3.3 and 3.4 (mappings of rankings to probabilities), as well as key design decisions in model formulation. Can you comment on sensitivity of the method to these values/decisions?

Will the evaluation data - and the code - be released?

372, 449: What instances lead to 'unknown' predictions for all factors? Is this a shortcoming with the factor generation method, or is it that the problem itself is riddled with uncertainty.

452: Can you elaborate on the spurious biases that affect CoT, giving it such strong performance? The same criticism could be applied to the LLM prompting as part of Bird, and the probabilities produced.

---

> ### Author Response · Authors · 2024-11-20
>
> We sincerely thank reviewer ZTvP for the constructive suggestions.  Here we address and clarify the questions you posed.
>
> **Paper writing and code release**
>
> We agree with the reviewer that the writing for the general setting could be more concise and that the method can be explained in greater detail, we plan to address these aspects more comprehensively in the final version as structural changes to the writing require a relatively large amount of time.
>
> We will first clarify some of your questions. All components within the central box, including the internal factors and assignments in the conditional probability tables for a given scenario, are automated and inferred through LLM interactions without knowing the specific downstream conditions. Later, the computations of observations (i.e., the condition-factor mapping) are also done automatically with LLMs. We include an extra instance in Fig. 2 that supports the opposite outcome, illustrating that BIRD can reliably compute probabilities and make decisions under any condition within the same scenario.
>
> We will release the complete codebase, including the code for all detailed methods of prompting the LLM, the learning algorithm, and the data.
>
>
> **Q1. Design choices of model parameters**
>
> We agree with the reviewer that the design choices for model parameters can be better elaborated. To address this, we have added detailed explanations of these design choices in Appendix A.3 and provide the same detailed explanation here.
>
> For most of the chosen values, we adopt a neutral and unbiased assumption. Consequently, we assign equal weights $w_j = 1, j = 1, ..., N$ and assume  $P(O_i) = 50\\%$ in Section 3.2. The value of $\mathbb{P}_{\rm init}(O_i|f_j)$ in Equation 5 Section 3.3  is assigned as it represents random initialization.
>
> To determine the mappings of rankings to probabilities in Section 3.3, we consulted two psychology experts and adopted the Likert scale theory. The finalized mapping between verbalized probabilities and numerical probabilities reflects typical human behavior and effectively distinguishes between verbalized probabilities in an unbiased manner. Additionally, slight adjustments to the mappings (~5%) do not noticeably affect overall performance.
>
> We randomly sample 128 instances from the LLM as training data to manage costs in Section 3.3. This represents the minimum number of instances required for effective training. Ideally, increasing the number of sampled instances would improve the model's alignment with the underlying LLM. All training hyperparameters for the algorithm are optimized using a grid search with a hold-out validation set from Plasma.
>
> We also conduct a preliminary ablation study on the loss function for decision making, as detailed below, and demonstrate that both components of the loss function are crucial for the model's performance.
>
> | Llama-3.1-70b-Instruct | w/o MR Loss | w/o MSE Loss | BIRD  |
> |-------------------------|-------------|--------------|-------|
> | TODAY                  | 65.5        | 65.2         | 74.3  |
> | Plasma                 | 60.0        | 65.7         | 73.0  |
> | Common2sense           | 77.2        | 86.8         | 92.3  |
>
>
>
> **Q3. 'Unknown' predictions**
>
> We verify that 98% of the unknown instances stem from shortcomings in the factor generation method, as these instances cannot be mapped to any factor value. While our framework is theoretically robust, challenges in implementation may arise due to the lack of comprehensiveness or sufficient granularity in the generated factors. To address this, we plan to explore and develop more effective factor generation methods in future work.
>
> At the same time, since BIRD does not rely on task gold labels to determine whether a prediction is unknown, we can leverage baseline methods (e.g., CoT) to generate predicted labels for these unknown cases. This approach ensures that our current comparisons, particularly regarding absolute performance gains, remain consistent even if we do not remove the unknown cases.
>
> **Q4. Spurious biases of LLMs**
>
> Language models operate as induction machines and thus may rely on spurious information about relevant concepts internally. This enables CoT-like inferences to make educated guesses that are correct in many cases but not necessarily for the right reasons to directly solve tasks. (as discussed in https://arxiv.org/pdf/2305.14825, https://arxiv.org/pdf/2311.09702, https://arxiv.org/pdf/2410.05229). BIRD provides an advantage by decomposing tasks into components with clearer individual objectives that leverage LLMs' strengths, such as abduction for factor generation, coarse estimation under complete information, and textual entailment. By structuring tasks in this way, each instance of LLM prompting becomes more controllable, mitigating the impact of spurious biases. Furthermore, BIRD incorporates a symbolic structure with explainable factors, encouraging more reliable and interpretable predictions.

---

> > ### Comment · Reviewer_ZTvP · 2024-11-25
> >
> > Thanks for your response, this is convincingly argued. I stand by my positive assessment.

---

### Official Review · Reviewer_gTw6 · 2024-11-02

**Soundness:** 2
**Presentation:** 3
**Contribution:** 3
**Rating:** 8
**Confidence:** 3

**Summary:**

This work proposes an inference framework that aims to produce more reliable and accurate probability/confidence estimations using LLMs for large-scale decision-making and planning tasks. The proposed method, BIRD, uses abductive factor generation to factorize the inference task, and estimates the outcome by accumulating the predictions made on the specific factors. The experimental results on three datasets show that BIRD is able to produce more accurate probability estimations in decision-making and planning, which can be used to (1) improve the accuracy of decision-making; (2) supervise the training of small models; (3) propose follow-up questions for decision-making. For all these applications, BIRD shows superior performance than the baseline methods.

**Strengths:**

1. The proposed Bayesian inference framework is conceptually interesting. Factoring the overall decision-making into multiple fine-grained factors is intuitive and useful for mitigating LLMs' limitations in providing accurate fine-grained probability/confidence estimation.

2. The experiments performed are carefully designed, and the overall writing is clear and with enough details.

3. The experiment results demonstrate the effectiveness of BIRD, showing its value in multiple application scenarios.

**Weaknesses:**

1. The training setting for "Learning algorithm for constrained optimization to estimate $P(O_i|f)$" (Line 266) should be more clearly described. The current description does not clarify (1) the number of learnable parameters ($P(O_i|f_j)$); (2) whether different instances share the same factors $f_j$; (3) whether there are $f_j$ in the test data that are unseen in the training data.

2. In the experiment for "Applying BIRD’s Probabilities in Decision-Making" (Line 430), the test instances on which BIRD's decision is unknown are filtered out. While it is claimed that "this does not undermine our experiment setting because such removal is label-agnostic" (Line 450), it may still introduce biases since this filtering process depends on the output of BIRD. It might be the case that BIRD predicts "unknown" when it is uncertain internally.

**Questions:**

In Section 4.3, "The Usage of the Reliably Estimated Probability", it seems that the original T5 large model is better on some datasets compared to the fine-tuned models. What would be the reasons for this? Besides, what is "PatternTime" in Table 3? It is not mentioned anywhere else in the manuscript.

Please see "Weaknesses" for a few additional questions.

---

> ### Author Response · Authors · 2024-11-20
>
> We sincerely thank reviewer gTw6 for the valuable comments. Here we answer all the questions you posted.
>
> **W1. The training setting for the learning algorithm**
>
> We agree with the reviewer that the training setting for the learning algorithm can be more clearly described. We have revised and added additional detailed explanations of the learning algorithm in Appendix A.3 and provide the same detailed explanation here.
> 1. For a given scenario $S$, the number of learnable parameters is the total number of values for all the generated factors, i.e., $\sum_{j=1}^{N} \text{card}(\mathcal{F}_j)$.
> 2. All training instances for a scenario are drawn from the complete information space for the same scenario $S$  and possible outcomes $\\{O\\}$ and will share the same factors  $\\{F_j\\}_{j=1}^N$.
> 3. LLMs generate the factors for a scenario without knowing the specific downstream conditions and their corresponding outcome labels, therefore the factors are the same in the process of fitting the Bayesian network and later inferences.
>
> **W2. Unknown instances**
>
> We verify that 98% of the unknown instances arise because these instances cannot be mapped to any factor value. While our framework is theoretically robust, challenges in implementation may stem from factor generation, as the generated factors may lack comprehensiveness or sufficient granularity. We plan to explore and develop more effective factor generation methods in future work to address this.
>
> At the same time, since BIRD does not rely on task gold labels to determine whether a prediction is unknown, we can leverage baseline methods (e.g., CoT) to generate predicted labels for these unknown cases. This approach ensures that our current comparisons, particularly regarding absolute performance gains, remain consistent even if we do not remove the unknown cases.
>
> **Q1.  Experiment settings for section 4.3**
>
> All experiment settings can be referenced in Feng et al. (2023)(https://aclanthology.org/2023.acl-long.671.pdf ), and we have also included them in Appendix A.9 in the updated version. The T5-large model and Patterntime  (Zhou et al., 2021)(https://arxiv.org/pdf/2010.12753) are specialized temporal reasoning models, as outlined in Feng et al. (2023). Both models were specifically fine-tuned on temporal reasoning datasets using the original temporal training data from Feng et al. (2023). In contrast, the additional training data used in our paper focuses on commonsense reasoning. Consequently, the additional finetuning aims to create a more general reasoning model, which may result in slightly reduced temporal reasoning performance. However, our primary objective is to demonstrate that finetuning with BIRD-produced probabilities yields better results than finetuning with hard classification labels. We show that using BIRD-produced probabilities preserves temporal reasoning capabilities while enhancing general reasoning abilities.

---

> > ### Comment · Reviewer_gTw6 · 2024-11-21
> >
> > Thank you for your detailed response. The response and updates to the paper have largely addressed my questions and concerns. I have updated my assessment accordingly.

---

### Meta-Review · Area_Chair_WGYo · 2024-12-23

**Metareview:**

The paper introduces BIRD, a Bayesian inference framework that combines LLMs with structured Bayesian networks to produce more reliable probability estimations. The method shows significant improvements in decision-making accuracy compared to LLM baselines and demonstrates applications in model supervision and follow-up question generation.

*Strengths*
Novel framework: BIRD uniquely combines LLM-based factor generation with Bayesian networks, addressing limitations in probability estimation.
Methodology: The framework effectively decomposes complex tasks into interpretable components while incorporating model uncertainty.
Empirical Validation: Thorough evaluation against strong baselines demonstrates consistent improvements across multiple datasets.

*Weaknesses*
-Some critical implementation details are deferred to appendices, such as LLM prompting methods.
-The framework produces "unknown" predictions for some instances (~2%) due to limitations in factor generation.
-While preliminary ablations show robustness, a more comprehensive analysis of parameter choices would strengthen the work.

All reviewers recommend acceptance, and recognize the paper's novel approach and strong empirical results. The authors have addressed concerns about technical details and unknown predictions in their responses.

**Additional Comments On Reviewer Discussion:**

The authors' response addresses most of the identified weaknesses

---

### Decision · Program_Chairs · 2025-01-22

Accept (Oral)